# Revised mineral dust emissions in the atmospheric chemistry-climate model EMAC (MESSy 2.52 DU_Astitha1 KKDU2017 patch)

Klaus Klingmüller[1], Swen Metzger[2], Mohamed Abdelkader[1,3], Vlassis A. Karydis[1], Georgiy L. Stenchikov[3], Andrea Pozzer[1], and Jos Lelieveld[1,2]

[1]Max Planck Institute for Chemistry, P.O. Box 3060, 55020 Mainz, Germany

[2]The Cyprus Institute, P.O. Box 27456, 1645 Nicosia, Cyprus

[3]King Abdullah University of Science and Technology, Thuwal 23955-6900, Saudi Arabia

*Correspondence to:* Klaus Klingmüller (k.klingmueller@mpic.de)

**Abstract.** To improve the aeolian dust budget calculations with the global ECHAM/MESSy atmospheric chemistry-climate model (EMAC) we have implemented new input data and updates of the emission scheme.

The data set comprises landcover classification, vegetation, clay fraction and topography. It is based on up-to-date observations, which is crucial to account for the rapid changes of deserts and semi-arid regions in recent decades. The new Moderate-resolution Imaging Spectroradiometer (MODIS) based landcover and vegetation data is time dependent, and the effect of long-term trends and variability of the relevant parameters is therefore considered by the emission scheme. All input data has a spatial resolution of at least $0.1°$ compared to $1°$ in the previous version, equipping the model for high resolution simulations.

We validate the updates by comparing the aerosol optical depth (AOD) at 550 nm wavelength from a one year simulation at T106 (about $1.1°$) resolution with Aerosol Robotic Network (AERONET) and MODIS observations, the 10 $\mu$m dust AOD (DAOD) with Infrared Atmospheric Sounding Interferometer (IASI) retrievals, and dust concentration and deposition results with observations from the AEROCOM dust benchmark data set. The update significantly improves agreement with the observations and is therefore recommended to be used in future simulations.

# 1 Introduction

Aeolian dust can impair everyday life and air quality especially in severe dust storms. Due to the worldwide presence of dust sources and through long range transport it has a significant global impact on atmospheric radiation transfer and air quality, affecting climate (IPCC, 2014) and human health (Giannadaki et al., 2014), which requires detailed representation in general circulation models (Shao et al., 2011).

Global models have different requirements regarding the dust emission scheme compared to regional models. As global models require planetary consistent input data sets, the availability of adequate data is more limited. Additionally, the coarser grid spacing requires an appropriate parametrisation of sub-grid processes, and, for example, reproducing individual dust events with global models may have lower priority than adequately representing the atmospheric dust budget on a longer time scale. But with their ever-increasing resolution, global models in many regards correspond to former generation regional models, and therefore established emission schemes are often applied in both, regional and global models.

Global models implement dust emissions with various complexity levels. Even the simplest version, prescribed (*offline*) dust emissions can produce acceptable results for the global aerosol distribution and variability due to the importance of atmospheric transport (Pozzer et al., 2015; Pringle et al., 2010b). Improved agreement with observations is generally achieved with *online* emission schemes which consider actual meteorological conditions, most importantly the surface friction velocity and the wind speed close to the surface. They are combined with a characterisation of surface properties, where properties and relations are to different degrees empirical (*source functions*) or deduced from micro-physical processes. The dominant processes considered are saltation bombardment by sand blasting and aggregate disintegration, and more elaborate emission schemes consider additional effects such as direct aerodynamic entrainment (Shao, 2001; Klose et al., 2014). The inability of most current global models to resolve convection means that haboobs which are responsible for a major fraction of the dust emissions (Marsham et al., 2013; Allen et al., 2013, 2015) are not represented at all. Therefore efforts are made to combine the emission schemes with explicit parametrisations of convective dust storms (Pantillon et al., 2015, 2016).

The global ECHAM/MESSy atmospheric chemistry-climate model (EMAC) (Jöckel et al., 2005, 2010) provides a choice of dust emission schemes (Tegen, 2002; Balkanski et al., 2004; Astitha et al., 2012) to calculate the emission flux online based on the meteorological conditions.

An advanced scheme producing convincing results when compared to observations has been presented by Astitha et al. (2012) building on previous studies (Pérez et al., 2006; Spyrou et al., 2010; Laurent et al., 2008, 2010; Marticorena et al., 1997; Zender et al., 2003; Tegen, 2002), and is the basis of the work presented here. Its basic principles are shared with emission schemes used in many other models (e.g. Zender et al., 2003; Jones et al., 2012; Albani et al., 2014; Huneeus et al., 2011), but alternative approaches exist (e.g. Shao, 2001; Kok et al., 2014). The emission scheme combines meteorological parameters with descriptions of landcover type, clay fraction of the soil and vegetation cover. One variant of the scheme (DU_Astitha2) additionally accounts for regional differences of the particle size distribution, while in the present study we focus on the simpler variant DU_Astitha1, which achieves competitive results with reduced complexity (Astitha et al., 2012)

and has proven to perform well in previous studies (Abdelkader et al., 2015, 2016). The emission scheme is summarised in appendix A.

The emission scheme applies physical principles in the sense that the governing equations are derived for microphysical processes that are consistently applied globally without the option to adjust the resulting emissions regionally. In this study we extend the emission scheme by including a topography factor while we strictly adhere to the global consistency concept and refrain from using regional tuning factors.

Though generally the original emission scheme produces convincing results, some shortcomings, predominantly related to the input data, have become apparent recently and are the motivation for the revision presented in this study. The original input data for land cover and vegetation is based on observations from the early 1990s and is thus dated in view of the rapid changes of deserts and semi-arid regions in recent decades (Figs. 1, 2, Klingmüller et al. (2016); Lamchin et al. (2016); Dong and Sutton (2015)). For instance, the emission mask resulting from the land cover data considerably limits emissions in the Middle East, essentially not allowing dust emissions in Syria and northern Iraq. This is in conflict with the emergence of severe dust outbreaks from that region (Solomos et al., 2017), and the strong link between the soil conditions in that region and trends of atmospheric dust over the Middle East (Klingmüller et al., 2016). Moreover, only a static land cover map and a single seasonal cycle for the vegetation index was provided.

As a consequence, the effect of variations and trends of these quantities on the modelled dust emissions have been excluded. Further, the resolution of the original input data is limited to $1°$. Particularly for EMAC simulations focusing on dust modelling, high model resolutions are desirable, considering how localised dust outbreaks can occur. In the long term, the resolution of global models will approach the resolution of today's regional models where high resolution input data are essential to include details of dust generation patterns (Shi et al., 2016; Anisimov et al., 2017). For model resolutions higher than T106 ($\approx 1.1°$) as applied in the present study, improved input data is required to justify the numerical effort. To equip the model for simulations at a resolution of T255 ($\approx 0.5°$) or higher, new input data should have at least $0.1°$ degree resolution.

In addition to updated input data addressing these issues, we present adjustments to the emission scheme to assure that the updated input has no undesirable effects such as too strong emissions in mountainous regions and to further improve the performance of the scheme.

To quantify the impact of the updates, we compare a validation simulation with the reference simulation, the latter using the original emission scheme and data. Results and comparisons of other schemes in EMAC are provided elsewhere (Gläser et al., 2012; Astitha et al., 2012). The purpose of the validation is to demonstrate the advantages of the updates and to test the results so that the modifications can swiftly be adopted by the community; more applications and in depth analysis thereof are beyond the scope of this mostly technical study.

The article is structured as follows: in Sect. 2 we introduce and discuss the updated input data; the modifications to the EMAC code are presented in Sect. 3 and their individual effects studied in Sec. 4. The effect of both is validated in Sect. 5 by comparing with the reference simulation, as well as ground based aerosol optical depth (AOD) observations (Sect. 5.1), and satellite based AOD (Sect. 5.2) and dust AOD (DAOD) (Sect. 5.3) retrievals as well as concentration and deposition data (Sect. 5.4).

## 2 Updated input data

### 2.1 Landcover

To replace the landcover classification map of Olson (1992), we use the MODIS MCD12C1 landcover product (MODIS MCD12C1) at $0.05°$ resolution, allowing for dust emissions from regions classified as *barren or sparsely vegetated*. Not only the resolution is higher than for the Olson data, which in the original emission scheme has been used at $1°$ latitude and longitude (aggregated from $10'$), but also yearly updated data from 2001 to 2012 are provided, also expecting more recent updates to become available. Therefore, changes of the landcover for example due to desertification are taken into account, which have not been considered previously. To assess these changes, we compute for each pixel the Kendall rank correlation coefficient $\tau$ of annual mask value, which can be either 0 (non-emitting) or 1 (emitting), and time; the result is shown in Fig. 1. Positive values of $\tau$ indicate an expansion of source regions to the respective pixel, negative values a disappearance of sources. In some regions the deserts are shrinking, e.g. in the Sahel, Central Asia and Australia. Expanding source areas are found rather centrally in the dust belt, e.g. in the Sahara, on both sides of the Red Sea and north of the Arabian Peninsula in Syria and Iraq. Globally, the area with positive correlation coefficients covers $1.3 \cdot 10^6$ km$^2$ which is about half the area with negative correlation coefficient $(2.6 \cdot 10^6$ km$^2)$. Additionally, the regions of shrinking deserts are spread over a larger area because unlike the centrally located expanding source regions they are predominantly surrounding the large deserts.

### 2.2 Vegetation

Yuan et al. (2011) have reprocessed the MODIS leaf area index (LAI) products to provide a temporally continuous and spatially consistent LAI data set for climate modelling that encompasses the time period since 2000. We have aggregated this data from 30" to $0.1°$ spatial resolution and from eight-day to one month temporal resolution. The data replaces the twelve month seasonal cycle of the vegetation area index with $1°$ resolution based on the work of Kergoat et al. (1999) and Bonan et al. (2002). Using continually updated monthly values instead of a repeating seasonal cycle implies that multi-annual vegetation trends are taken into account.

The LAI data is used to compute the vegetation factor (Astitha et al., 2012),

$$f_{\text{veg}} = 1 - \frac{\min(\text{LAI}, 0.35)}{0.35}. \tag{1}$$

which linearly interpolates between full emissions for no vegetation and entirely suppressed emissions for LAI $\geq 0.35$ which was introduced as threshold by Mahowald et al. (1999). The 16 year average, standard deviation of the yearly averages and the trend of the vegetation factor are shown in Fig. 2. The trend has been calculated as slope of a linear regression model fitted to the annual averages using least squares; only pixels with p values below the significance level of 0.05 are plotted. As demonstrated by the standard deviation plot, large variability and trends, e.g. related to changing desert boundaries, coincident with the regions of landcover changes, as shown in Fig. 1 can strongly influence the results. The strongest variability is observed in the interior lowlands of Australia (Simpson, Strzelecki and Tirari Deserts), the Thar Desert (India/Pakistan) and Mesopotamia.

While in Australia the variability does not yield a significant trend over the 16 year period, in and around the Thar desert a strong decrease of the vegetation factor, indicating vegetation growth, is observed. This inhibits dust emissions and could result in the significant negative AOD trend in that region reported by Klingmüller et al. (2016). In contrast, vegetation decreases in Syria and Iraq, resulting in a larger vegetation factor and more dust emissions. However, similar to Australia, considering the strong variability, the trend is not very distinct because the highest vegetation factor in Iraq and Syria occurred in 2008 in the middle of the period of available data, whereas it decreased again in recent years.

## 2.3 Clay fraction

The efficiency of the sandblasting process is very sensitive to the clay fraction of the surface soil. Both very small and very large clay fractions are assumed to suppress the sandblasting efficiency. Our parametrisation of this dependency is discussed in section 3. Replacing the $1°$ clay fraction map of Scholes and Brown de Colstoun (2011), here we employ higher resolved clay fraction data from the *Global Soil Dataset for use in Earth System Models* (GSDE) (Shangguan et al., 2014), aggregated from 30" to $0.1°$. The GSDE provides the clay fraction of the topmost 4.5 cm soil layer, which is most relevant for sandblasting rather than the clay fraction of the topmost 30 cm in the data of Scholes and Brown de Colstoun (2011). The two datasets are compared in Fig. S1 in the supplement.

## 3 Modifications to the emission scheme

*Sandblasting efficiency:* The sandblasting efficiency used by Astitha et al. (2012), based on the studies of Marticorena and Bergametti (1995) and Tegen (2002), increases exponentially with a clay fraction up to 20 %, beyond which the sandblasting is negligible, see Fig. 3. The resulting threshold is problematic in regions where the clay fraction is in the range of this discontinuity, for example in Iraq and Syria: small variations in the clay fraction can drastically alter the sandblasting efficiency between its maximum and essentially zero. Considering that both the clay fraction data and the sandblasting efficiency measurements are associated with uncertainty, we propose to apply a Gaussian filter. Figure 3 shows the efficiency after applying a filter with an interquartile range of 5 %, which is used in the validation simulation discussed below. The filter width could be optimised systematically, but in our experience results are robust by smoothing the distinct peak at 20 % clay fraction. Combining the filtered sandblasting efficiency with the updated clay fraction data (section 2.3) yields the global map presented in Fig. 4.

*Soil moisture term:* The original emission scheme of Astitha et al. (2012) applies a soil moisture dependent correction factor to the threshold friction velocity which increases the threshold and thus reduces dust emissions from wet soils. This correction factor has not been active in MESSy versions up to 2.52 and the higher AOD over the Middle East obtained without the factor generally resembles the satellite observations more closely, its impact when evaluated using soil moisture values from the current EMAC bucket model is rather small (see Fig. S2 in the supplement). Therefore, it remains inactive for the present study, consistent with previous studies (Abdelkader et al., 2015, 2016; Metzger et al., 2016; Albani et al., 2014). Nevertheless, the monthly vegetation data described above accounts for secondary effects of soil moisture variations via the vegetation factor. However, since the soil moisture is known to be a relevant parameter (Gherboudj et al., 2015) and, e.g., strongly correlates with

the AOD over the Middle East (Klingmüller et al., 2016) suggesting a direct link between surface drying and increasing dust emissions, we consider a detailed parametrisation of the soil moisture effect to be essential to capture the observed trends in future simulations. This will require a comprehensive soil model providing accurate moisture values for the topmost surface layer which has yet to be implemented in EMAC.

*Surface friction velocity limit:* The relation of the horizontal dust particle flux $H$ and the surface friction velocity $u_*$ is parametrised as a polynomial of degree 3,

$$
H \propto \begin{cases} (u_* + u_{*\mathrm{t}})^2(u_* - u_{*\mathrm{t}}) & u_* > u_{*\mathrm{t}} \\ 0 \end{cases}
\tag{2}
$$

where $u_{*\mathrm{t}}$ is the threshold friction-velocity. Therefore, high surface friction velocities occurring in mountainous regions can produce spuriously strong dust outbreaks where emissions are not limited by the updated landcover mask, vegetation factor or

sandblasting efficiency, e.g. in Iran. To avoid this, we limit the friction velocity in the above equation to a maximum value of 0.4 m / s. Figure S3 in the supplement exemplifies the effect of using larger or smaller limits. The precise limit might be further adjusted but the given value yields good results as shown in Sect. 5.

   *Topography factor:* In the original scheme, the accumulation of sediments in valleys and depressions is not considered explicitly and is only to some extent reflected implicitly by other input data such as the clay fraction. As shown by the reference

simulation presented in Sect. 5, this can result in an underestimation of dust emissions from areas like the Tigris-Euphrates Basin. We therefore include a topography factor using the topographic source function proposed by Ginoux et al. (2001),

$$
S_{\mathrm{topo}} = \left( \frac{z_{\max} - z}{z_{\max} - z_{\min}} \right)^5,
\tag{3}
$$

where $z$ is the median elevation in a circle with $1°$ diameter and $z_{\min}$ ($z_{\max}$) the minimum (maximum) elevation in the surrounding circle with $10°$ diameter. (Ginoux et al. (2001) use $1°$ pixels and the extreme values in the surrounding $10° \times 10°$

square). The Global Multi-resolution Terrain Elevation Data 2010 (GMTED2010) (Danielson and Gesch, 2011; GMTED2010, 2010) is used as topography data base. Figure 5 depicts a global map of the resulting topography factor. As the topography factor takes values between 0 and 1 and usually is smaller than 1, a normalisation factor $N \geq 1$ has to be multiplied to avoid suppression of the global emissions. In a one-month test simulation we obtain a ratio between the global emissions without and including the factor $S_{\mathrm{topo}}$ of 5.3. Consequently, the full topography term we use is $N S_{\mathrm{topo}}$ where $N = 5.3$.

*Mode mapping:* The emission scheme considers emissions into three log-normal modes, adapting the parameters of the "background" modes of d'Almeida (1987) listed in table 2. Originally, these log-normal modes have been mapped to eight transport bins as used by Pérez et al. (2006), before being distributed to the accumulation and coarse mode of the EMAC aerosol submodel GMXE. We simplify this procedure by directly mapping the three emission modes to the two relevant

GMXE modes. The mass fraction $M$ assigned to each GMXE mode is

$$M = \sum_{i=1}^{3} \frac{1}{2} \left( \mathrm{erf}(\frac{\ln(d_{\max}/\tilde{d}_i)}{\sqrt{2}\ln\sigma_{g,i}}) - \mathrm{erf}(\frac{\ln(d_{\min}/\tilde{d}_i)}{\sqrt{2}\ln\sigma_{g,i}}) \right), \tag{4}$$

where the sum encompasses over the three emission modes, $\tilde{d}_i$ and $\sigma_{g,i}$ are the mass median diameter and geometric standard deviation of each emission mode, and $d_{\min}$ and $d_{\max}$ are the threshold diameters of the GMXE mode. In practice, the modification is equivalent to a change of the threshold diameter between accumulation and coarse mode, which is now consistent with the GMXE parameters. Moreover, the algorithm generalises seamlessly when including additional GMXE modes such as a giant aerosol mode ($> 10\mu$m).

*Scaling factor:* For the dimensionless empirical constant $c$ by which the horizontal particle flux is scaled, Astitha et al. (2012) use the value $c = 1$, consistent with Darmenova et al. (2009). Since the dust emissions, especially in the Middled East, tend to underestimate the observations, we increase the value to $c = 1.5$, which is bounded by the original value and $c = 2.61$ used by White (1979) and Marticorena and Bergametti (1995). When switching to different model resolutions, the scaling factor can be used to balance potential resolution dependencies of the emission scheme. As will be discussed in section 5, with this value we obtain the same total amount of globally emitted dust as with the original emission scheme by Astitha et al. (2012). It should be stressed that the scaling factor is the central empirical tuning parameter of the emission scheme and might be improved by systematic optimisation, but our focus is on the spatiotemporal emission pattern which is largely unaffected by the overall scaling.

*Chemical composition:* In addition to the bulk dust flux output, we compute the $Na^+$, $K^+$, $Ca^{++}$ and $Mg^{++}$ fractions of the emitted dust, since mineral cations are important for the gas-aerosol partitioning (Metzger et al., 2006). For this purpose we have generated maps of the desert soil composition (Fig. 6) based on the fractions reported by Karydis et al. (2016) and geographical data from the Natural Earth dataset (Natural Earth, 2016). The chemical composition does not affect the amount of dust emitted, but the chemical ageing of airborne dust particles simulated by the GMXE submodel can affect the atmospheric residence time (Abdelkader et al., 2015) and the optical properties (Klingmüller et al., 2014).

## 4   Effects of the individual modifications

To compare the effects of the individual modifications we study the term $a\, f_{\mathrm{landcover}}\, f_{\mathrm{veg}}\, N\, S_{\mathrm{topo}}$ (cf. Eq. (A2) in the appendix), the product of the clay fraction dependent sandblasting efficiency $a$, the barren land fraction $f_{\mathrm{landcover}}$, the vegetation factor $f_{\mathrm{veg}}$ and the normalised topography factor $N\, S_{\mathrm{topo}}$. It is proportional to the dust emission flux (given that the threshold surface friction velocity is exceeded) and reflects the effects of the modifications independently of the precise wind conditions. Figure 7 compares the term during July 2011 for the reference and validation simulations, and variations of the validation setup selectively using either the landcover, sandblasting efficiency, clay fraction or vegetation data from the reference scheme, or omitting the topography factor. The update of the landcover data, the inclusion of the topography factor and the modification to the sandblasting efficiency distinctively affect the dust emissions, whereas the update of clay fraction and vegetation data

have a more subtle effect (see also Fig. S4 in the supplement). The latter implies that the effect of the seasonal cycle in the vegetation data is not clearly visible in this representation, justifying to study only July in Fig. 7. The landcover update clearly expands the source regions of the dust belt. The topography factor redistributes the emissions enhancing emissions from basins (e.g., the Tigris-Euphrates Basin) while reducing emissions from mountenous areas. Omitting the topography factor the revised scheme produces a much more homogeneous distribution. The revised sandblasting efficiency avoids pixels with very strong or very little emissions in regions with a clay fraction of around 20 %. In such regions, reverting to the original sandblasting efficiency yields peaks of extremely high emission factors, defining the upper limit of the colour scale in Fig. 7. This is especially the case in regions where the original scheme suppressed emissions based on the landcover classification, therefore the revised sandblasting efficiency is mandatory when using the updated landcover data. Most importantly, to the benefit of future high resolution simulations with truncations of T255 or higher ($< 50$km grid spacing), the updates considerably increase the resolution of the emission factor as illustrated by the column on the right hand side of Fig. 7.

## 5 Validation

We use EMAC in the combination ECHAM 5.3.02 and MESSy 2.52 at horizontal resolution T106 with 31 vertical levels. The Gaussian T106 grid has a grid spacing of $1.125°$ along the latitudes and about $1.121°$ along the longitudes. At the equator, this corresponds to virtually quadratical cells with around 125 km edge length. The following MESSy submodels have been enabled: AEROPT, AIRSEA, CLOUD, CLOUDOPT, CONVECT, CVTRANS, DDEP, GMXE, JVAL, LNOX, MECCA, OFFEMIS, ONEMIS, ORBIT, ORACLE, PTRAC, RAD, SCAV, SEDI, SURFACE, TNUDGE, TROPOP. Descriptions of each submodel and further references can be found online in the MESSy submodel list (MESSy 2017). The dust emission scheme is evaluated by the online emission submodel ONEMIS, the aerosol microphysical processes are simulated by the Global Model aerosol eXtEnsion (GMXE) submodel (Pringle et al., 2010a, b). Within GMXE two gas-aerosol partitioning schemes are available, ISORROPIA II (Fountoukis and Nenes, 2007) and EQSAM4clim (Metzger et al., 2016), here we employ the former. The prognostic radiative-transfer calculation uses the Tanre aerosol climatology for extinction, single scattering albedo and asymmetry factor (Tanre et al., 1984), and the model dynamics above the boundary layer are nudged to meteorological analyses of the European Centre for Medium-Range Weather Forecasts (ECMWF). The CMIP5 (Coupled Model Intercomparison Project), GFEDv3.1 (Global Fire Emissions Database) and AeroCom (Aerosol Comparisons between Observations and Models) databases provide anthropogenic, biomass burning and sea salt emissions, respectively.

Two simulations are considered: a reference simulation using the original emission scheme and a validation simulation using the updated input data presented in Sect. 2 and the modifications presented in Sect. 3. The different input datasets are summarised in table 1. The chemical composition of the emitted particles is considered in both simulations. As validation time period we selected the year 2011 to represent a recent period well past the time period on which the former input data was based on. The simulations are initialised at 1 July 2010 from the output of a lower resolving T42 simulation starting in 1998. After this initialisation, six months simulated with the final T106 resolution serve as additional spin-up period.

To quantify the (dis)agreement of model results and observations we use the skill score $S$ defined by Taylor (2001),

$$S = \frac{4(1+r)^4}{(\sigma_1/\sigma_2 + \sigma_2/\sigma_1)^2(1+r_0)^4}, \tag{5}$$

where $r$ is the correlation coefficient and $\sigma_1$ and $\sigma_2$ are the standard deviations of modelled and observed values. As maximum attainable correlation coefficient we simply use $r_0 = 1$ since we are predominantly interested in the relative changes of the skill score resulting from our modifications to the dust emission scheme. A more accurate estimate $r_0 < 1$ would result in higher skill scores.

Both simulations obtain the same global mineral dust emission of 1.3 Gt in 2011 (Table 3), which is well in the range of values reported by Huneeus et al. (2011) and close to their median of 1.1 Gt per year. Aligning the threshold between accumulation and coarse mode with GMXE as described in section 3 for the parameters shown in Table 2 results in more accumulation mode emissions in the validation simulation (0.15 Gt / year) than in the reference simulation (0.052 Gt / year), thus higher 550 nm AOD values are expected in the former.

## 5.1 AERONET

For the comparison with Aerosol Robotic Network (AERONET) (Holben et al., 1998; AERONET) AOD observations, we select regions based on the relevance of the regional dust emissions and the abundance of AERONET stations. We focus on the seven regions of interest depicted in Fig. 8 encompassing the Middle East (*ME*), north-west Africa (*N. Afr.*), Africa, Central and East Asia (*Asia*), the south-west of the United States of America (*N. Amer.*), the Southern Cone in South America (*S. A.*) and Australia (*Austral.*). All stations with observations during at least 120 days distributed over at least 9 months of 2011 are considered.

We compare daily averages of modelled and observed aerosol optical depth (AOD) at 550 nm, where the AERONET AOD at this wavelength is obtained from level 2 data by interpolation using the Ångström exponent. For each station we use the model values from the grid cell covering the station coordinates. The skill score $S$ is shown in Fig. 9. For most stations, the validation simulation achieves higher skill scores than the reference simulation (time series plots for the stations with the highest increase are shown in Fig. S5 in the supplement), similar skill scores are obtained for the Australian stations. Only over four stations in north-west Africa the validation simulations produces noticeably lower skill scores than the reference run. However, the skill scores for these stations remain among the highest globally. Moreover, the two stations with the strongest skill score degradation are located very close to each other on the island Tenerife, in Santa Cruz de Tenerife and at the Izana Atmospheric Observatory on Mount Teide. In contrast, the validation skill score for a third station on Tenerife, in La Laguna, is marginally larger than the corresponding reference skill score.

Studying the AOD time series for these three stations (Fig. 10 top), reveals that over Santa Cruz de Tenerife the model slightly overestimates the observations and the even higher AOD levels in the validation simulation result in the lower skill score. On the other hand, dust events observed by AERONET in January and December are reproduced by the validation simulation, but not by the reference simulation. The Izana station on Mount Teide is special: located at 2391 m altitude, it

shares the same model grid cell with the La Laguna station at 568 m altitude, Fig. 10 (bottom), but naturally the observed AOD is much lower. Obviously, the station site is not well represented by the model grid cell, which predominantly covers open sea and has a surface altitude of 63 m. These considerations put the regression of the skill score over the Canaries into perspective and suggest that some overestimation of the AOD over north-west Africa in the validation simulation is an acceptable trade-off

in view of the skill score increase elsewhere. This conclusion is further supported by the comparison with MODIS observations in the following section.

## 5.2   MODIS

To verify the global aerosol distribution, we validate the model AOD against observations from the Terra satellite provided by the Moderate-resolution Imaging Spectroradiometer (MODIS) data collection 6 (Hubanks et al., 2015; Levy et al., 2013;

MODIS MOD08 M3). We use the merged 550 nm AOD combining retrievals from the Deep Blue and Dark Target algorithms (Sayer et al., 2014).

Figure 11 compares the 2011 annual mean AOD from the two simulations and MODIS. The AOD levels over the Sahara and the Middle East produced by the validation simulation agree well with the observed levels, whereas they are underestimated by the reference simulation. Features of the MODIS distribution found in the validation but not in the reference result are

regionally high AOD values over the Middle East along the Gulf and extending over Iraq and Syria, and the absence of a local maximum over Argentina. The latter is even more evident at higher wavelengths considered in the following section. Over west Africa, the high AOD levels in the validation simulation extend slightly further north than observed by MODIS. This is consistent with the overestimation of AERONET observations in that region discussed above, but does not considerably compromise the globally improved agreement with MODIS.

The improved agreement of the AOD distribution obtained by the validation simulation can be quantified by correlating the pixel values of the equivalent maps shown in Fig. 11. The revised dust emissions enhance the spatial correlation of the AOD pattern from 0.79 to 0.81 and the skill score from 0.58 to 0.67.

Fig. 12 zooms into the Middle East (Region A) to illustrate the annual variability of the 550 nm AOD by showing seasonal means. Especially in Spring and Summer, the enhanced AOD levels along the Tigris-Euphrates Basin and the Gulf are clearly

visible in the validation result, consistent with the MODIS observations, while not being represented in the reference results. During summer, the validation simulation produces higher AOD levels also over Arabian and Red Sea, which are closer to the extremely high levels reported by MODIS and Brindley et al. (2015). Surprisingly, the MODIS AOD over Iran is close to zero throughout the year, but substantial levels are obtained during spring and summer by both simulations, with higher levels in the validation simulation than in the reference simulation. The strong seasonal cycle over the Middle East observed by MODIS is

reproduced by both simulations, but with its higher spring and summer AOD levels, the validation simulation yields a higher amplitude in better agreement with MODIS. To underscore the improvement achieved by the revised emissions, we quantify the spatial agreement of the seasonal AOD over the Arabian Peninsula including Syria, Iraq and Jordan using the correlation coefficient and the skill score (see Fig. S6 in the supplement). Both measures show a significant increase throughout the year, especially during winter (the correlation coefficient from 0.18 to 0.54, the skill score from 0.068 to 0.24) and summer (the

correlation coefficient from 0.46 to 0.75, the skill score from 0.22 to 0.55). The global seasonal AOD distribution is shown in Figs. S8 to S11 in the supplement.

## 5.3 IASI

To focus the evaluation more tightly on dust, we utilise data from the *Infrared Atmospheric Sounding Interferometer* (IASI)
(Clerbaux et al., 2009; Hilton et al., 2012) provided by the Aerosol-CCI (Climate Change Initiative) project (Popp et al., 2016; IASI) of the *European Space Agency* (ESA). We use version 7 of the level 3 monthly dust AOD (DAOD) at 10 $\mu$m prepared at the Université Libre de Bruxelles (IASI_ULB.v7). The corresponding annual average DAOD map for 2011 is shown in the middle panel of Fig. 13.

To compare with the IASI DAOD, we filter the daily 10 $\mu$m EMAC AOD considering only dust dominated values as DAOD,
setting the DAOD to zero if sea salt dominates instead. The contribution of both components is quantified by weighting the AOD of each mode with the volume fraction of the component. The diagnostic output of optical properties at wavelengths up to 10 $\mu$m has not been utilised previously in EMAC though proves very valuable to compare with remotely sensed optical properties of coarse particles such as aeolian dust. The annual average for 2011 from validation and reference simulation are shown in the top and bottom panel of Fig. 13. In several aspects the DAOD distribution obtained by the validation simulation resembles
the IASI observations more closely. In the Middle East, the region of high dust loads distinctly extends north-westwards into the Fertile Crescent, whereas comparably low dust loads are found over the western half of the Arabian Peninsula. The DAOD is more pronounced over Pakistan, and similarly over Djibouti and the adjacent regions south-west of the Red Sea. The regional maximum over Chad is less distinct than in the reference simulation. Over the Southern Andes, the maximum obtained by the reference simulation, though not detected by IASI, is not reproduced by the validation simulation, which is distinctly more
realistic.

The correlation coefficient of the validation result and IASI is 0.89 compared to 0.79 for the reference simulation, the corresponding skill score is enhanced by our modifications from 0.64 to 0.78.

The annual variability of the 10 $\mu$m DAOD over the Middle East (Region A) is compared in Fig. 14. As for the AOD, in spring and summer, the high DAOD values along the Tigris-Euphrates Basin are clearly visible in the validation result, consistent with
the IASI observations, while not being represented in the reference result. During summer, the DAOD pattern obtained by the validation simulation at the southern Red Sea resembles the pattern observed by IASI, even though the observed regional maximum is more pronounced. Also the DAOD at the Iranian and Pakistani Arabian Sea coast produced by the validation simulation agrees more closely with the IASI result. The reference simulation does not produce dust over the Caspian Sea and to its south, whereas IASI obtains significant DAOD values in spring and summer. These are reproduced by the validation
simulation but seem to be slightly overestimated during summer. The strong seasonal cycle observed by IASI is realistically reproduced by both simulations. We quantify the apparent improvement achieved by the revised emissions by assessing the spatial agreement of the seasonal AOD over the Arabian Peninsula (including Syria, Iraq and Jordan) using the correlation coefficient and the skill score (see Fig. S7 in the supplement). The increase obtained for both measures throughout the year

is significant for most seasons, especially during autumn for which the correlation coefficient increases from 0.30 to 0.62, the skill score from 0.14 to 0.39. The global seasonal DAOD distribution is shown in Figs. S12 to S15 in the supplement.

## 5.4 Dust concentration and deposition

We use dust concentration and deposition data from the AEROCOM dust benchmark dataset (Huneeus et al., 2011) to evaluate the corresponding results of our simulations. Concentration climatologies from 25 sites with in total 292 monthly values and the annual dust deposition rates from 84 sites are considered for our evaluation (see Fig. S16 in the supplement).

The deposition obtained by the validation simulation agrees significantly better with the observations than the reference result (Fig. 15), with a correlation coefficient of 0.89 compared to 0.80 and a skill score of 0.78 compared to 0.64. Regarding the concentration, the two simulations show no significant difference in performance.

At sites with low dust concentrations both simulations underestimate the observed concentrations which could be either due to an underestimation of dust transport in the model or due to local non-desert dust sources not represented in the dust emission schemes.

## 6 Conclusions

We have prepared new input data for use with the EMAC dust emission scheme developed by Astitha et al. (2012), and proposed changes and extensions. With a geographic representation of at least $0.1°$ for all input parameters, the updated input data has a significantly higher spatial resolution than the data used thus far. Therefore, the new data will be important for use in planned high resolution simulations with truncations of T255 or higher ($< 50$km). The land cover and vegetation in the updated data is time dependent, so that the effect of long-term trends and variability of these quantities on the dust emissions are taken into account. In addition to the input parameters used by the original implementation by Astitha et al. (2012), we take the topography into account, which enhances the emissions from basins and valleys such as the Tigris-Euphrates region and the Afar Triangle, in better agreement with observations. Moreover, we have produced soil composition maps to differentiate the chemical composition of dust particles from different deserts that affects the coating of mineral dust by hygroscopic salts during atmospheric ageing.

The updated landcover classification, the inclusion of the topography factor and the modification of the sandblasting efficiency function have a considerable impact on the global and regional distribution of dust emissions. By comparison, the effect of the clay fraction and vegetation data updates is less distinct.

The updated input data in combination with the adjustments to the emission scheme improve the modelled AOD and DAOD, as demonstrated by the comparison with AERONET, MODIS and IASI observations. For this validation, we have evaluated the EMAC DAOD at wavelengths up to $10~\mu$m for the first time, which allows testing of the model with a focus on dust, i.e. based on IASI DAOD.

Also the comparison with dust deposition observations shows improved agreement when using the updated emissions. This is less clear for the comparison with dust concentration data, where original and updated emission scheme do not show a significant performance difference.

While the updates clearly improve the global distribution of aeolian dust, the total amount of globally emitted dust remains unchanged and consistent with literature values.

Subject to the future availability of suitable soil models in EMAC providing soil moisture values for a thin surface soil layer, the activation of the explicit soil moisture dependency of the threshold surface friction velocity might further improve the agreement with observed trends and variability.

**Code and data availability**

The Modular Earth Submodel System (MESSy) is continuously further developed and applied by a consortium of institutions. The usage of MESSy and access to the source code is licenced to all affiliates of institutions which are members of the MESSy Consortium. Institutions can become a member of the MESSy Consortium by signing the MESSy Memorandum of Understanding. More information can be found on the MESSy Consortium Website (http://www.messy-interface.org). The input data files and all modifications to the EMAC source code presented in this article are available on request until they become part of the official MESSy code.

*Acknowledgements.* The research reported in this publication has received funding from the King Abdullah University of Science and Technology (KAUST) CRG3 grant URF/1/2180-01 *Combined Radiative and Air Quality Effects of Anthropogenic Air Pollution and Dust over the Arabian Peninsula*. S. Metzger receives funding from the European Commission through the H2020-EINFRA-2015-1 project "Energy oriented Centre of Excellence for computer applications (EoCoE)", Proposal number: 676629.

## Appendix A: Emission equation

In the DU_Astitha1 emission scheme (Astitha et al., 2012), the threshold surface friction velocity $u_{*t}$ is obtained by the equation

$$
\begin{aligned}
u_{*t} = {} & 0.129 \sqrt{\frac{D_{\mathrm{p}}}{\rho_{\mathrm{air}}} \left( \rho_{\mathrm{p}} g + \frac{0.006 \mathrm{g} \sqrt{\mathrm{cm}}/\mathrm{s}^2}{D_{\mathrm{p}}^{5/2}} \right)} \\
& \times \begin{cases} \dfrac{1}{\sqrt{1.928 B^{0.092} - 1}} & B < 10 \\ (1 - 0.0858 e^{-0.0617(B-10)}) & B \geq 10 \end{cases} \\
& \times \left( 1 - \frac{\ln \frac{z_{\mathrm{o}}}{z_{\mathrm{os}}}}{\ln(0.35 \left( \frac{10\mathrm{cm}}{z_{\mathrm{os}}} \right)^{0.8})} \right)^{-1} \\
& \times \sqrt{1 + 1.21 \max(0, \left( w - (0.0014 \phi_{\mathrm{clay}}^2 + 0.17 \phi_{\mathrm{clay}}) \right))^{0.68}},
\end{aligned}
\tag{A1}
$$

where

| | |
|---|---|
| $D_{\mathrm{p}} = 60\ \mu m$ | saltation particle diameter |
| $\rho_{\mathrm{air}}$ | air density |
| $\rho_{\mathrm{p}} = 2.65\ \mathrm{g/cm}^3$ | particle density |
| $g = 9.80665\ \mathrm{m/s}^2$ | gravitational acceleration |
| $B = \frac{u_{*t} D_{\mathrm{p}}}{v}$ | friction Reynolds number, |
| | initially $B = 1331(D_{\mathrm{p}}/\mathrm{cm})^{1.56} + 0.38$ |
| $v = 0.157 \cdot 10^{-4}\ \mathrm{m}^2/\mathrm{s}$ | kinematic viscosity of air |
| $z_{\mathrm{o}} = 0.01\ \mathrm{cm}$ | surface roughness length |
| $z_{\mathrm{os}} = 0.00333\ \mathrm{cm}$ | local roughness length of the uncovered surface |
| $w$ | gravimetric soil moisture in % |
| $\phi_{\mathrm{clay}}$ | clay fraction in % |

The last, soil moisture term in Eq. (A1) is omitted in the present study. If the surface friction velocity $u_*$ exceeds the threshold $u_{*t}$, the resulting emission flux is computed according to the equation

$$
j_{\mathrm{emis},i} = \frac{c \rho_{\mathrm{air}}}{g} (u_* + u_{*t})^2 (u_* - u_{*t})\ 10^{-4}\ a\ f_{\mathrm{landcover}}\ f_{\mathrm{veg}} M_i,
\tag{A2}
$$

where

| | |
|---|---|
| $i$ | mode index |
| $c = 1$ | empirical constant (in this study $c = 1.5$) |
| $u_*$ | surface friction velocity |
| $f_{\text{landcover}}$ | barren land fraction |
| $f_{\text{veg}} = 1 - \frac{\min(\text{LAI}, 0.35)}{0.35}$ | vegetation factor |
| $a$ | sandblasting efficiency |
| $M_i$ | mass fraction emitted into mode $i$ |

In the present study we multiply the right-hand side of Eq. (A2) with the topography factor $S_{\text{topo}} = ((z_{\text{max}} - z)/(z_{\text{max}} - z_{\text{min}}))^5$ defined in Eq. (3) and the corresponding normalisation factor $N = 5.3$. In addition, the surface friction velocity $u_*$ is limited to a maximal valure of $0.4 \, \text{m/s}$, i.e., $u_*$ in Eq. (A2) is replaced by $\min(u_*, 0.4 \, \text{m/s})$.

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

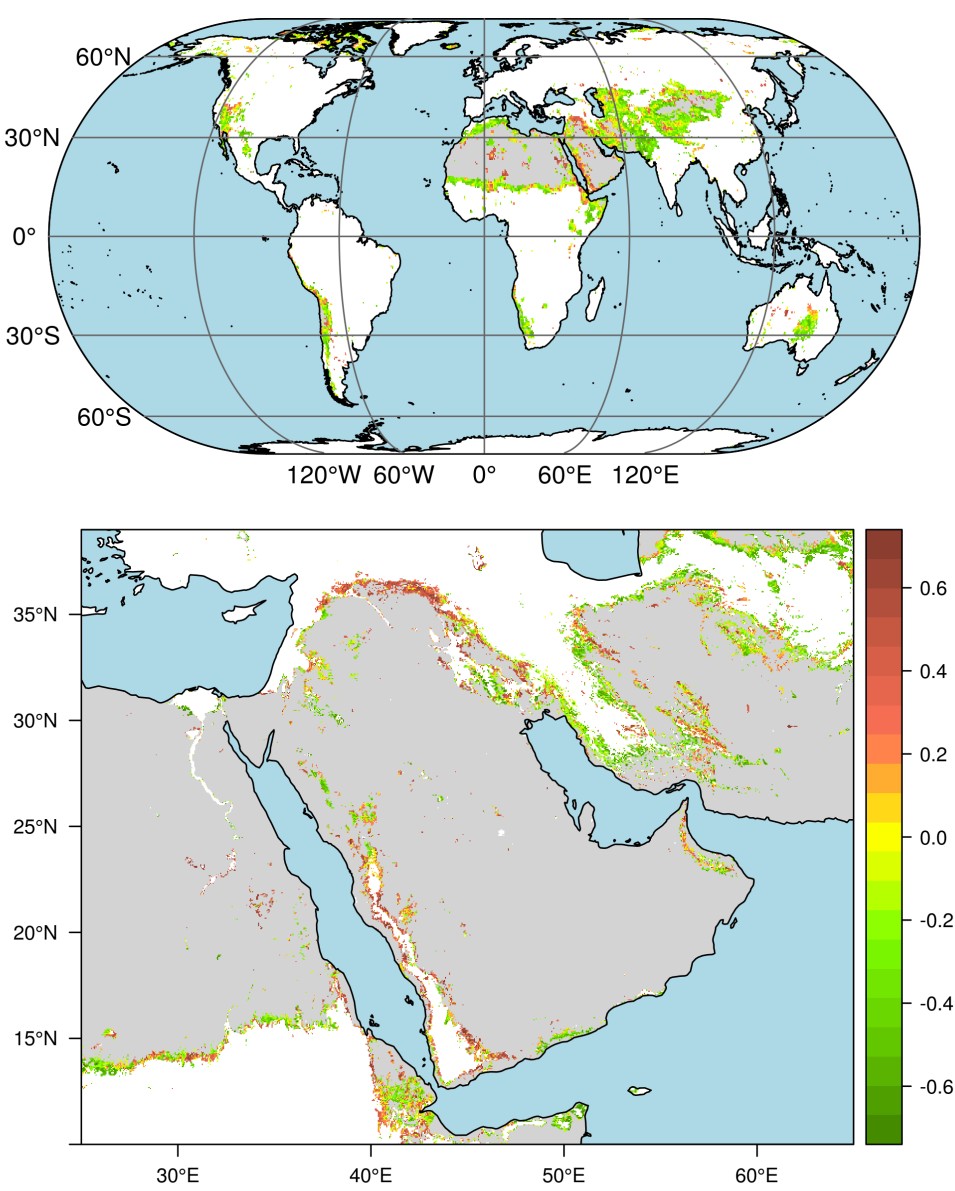

**Figure 1.** Trend of the dust emission mask based on the MODIS MCD12C1 landcover product during the period 2001 to 2012. Regions with changing surface properties are coloured according to the Kendall rank correlation coefficient $\tau$ of time and mask value, depicting expansion of source regions (i.e., positive correlation coefficients) in red, and contraction in green. Regions where the land cover remained unchanged are grey (source regions) or white (non-source regions). For better readability, in the global plot (top) the values have been averaged over 10 by 10 pixels ignoring constant pixels. The magnified plot of the Middle East (bottom) shows the original $0.05°$ pixels.

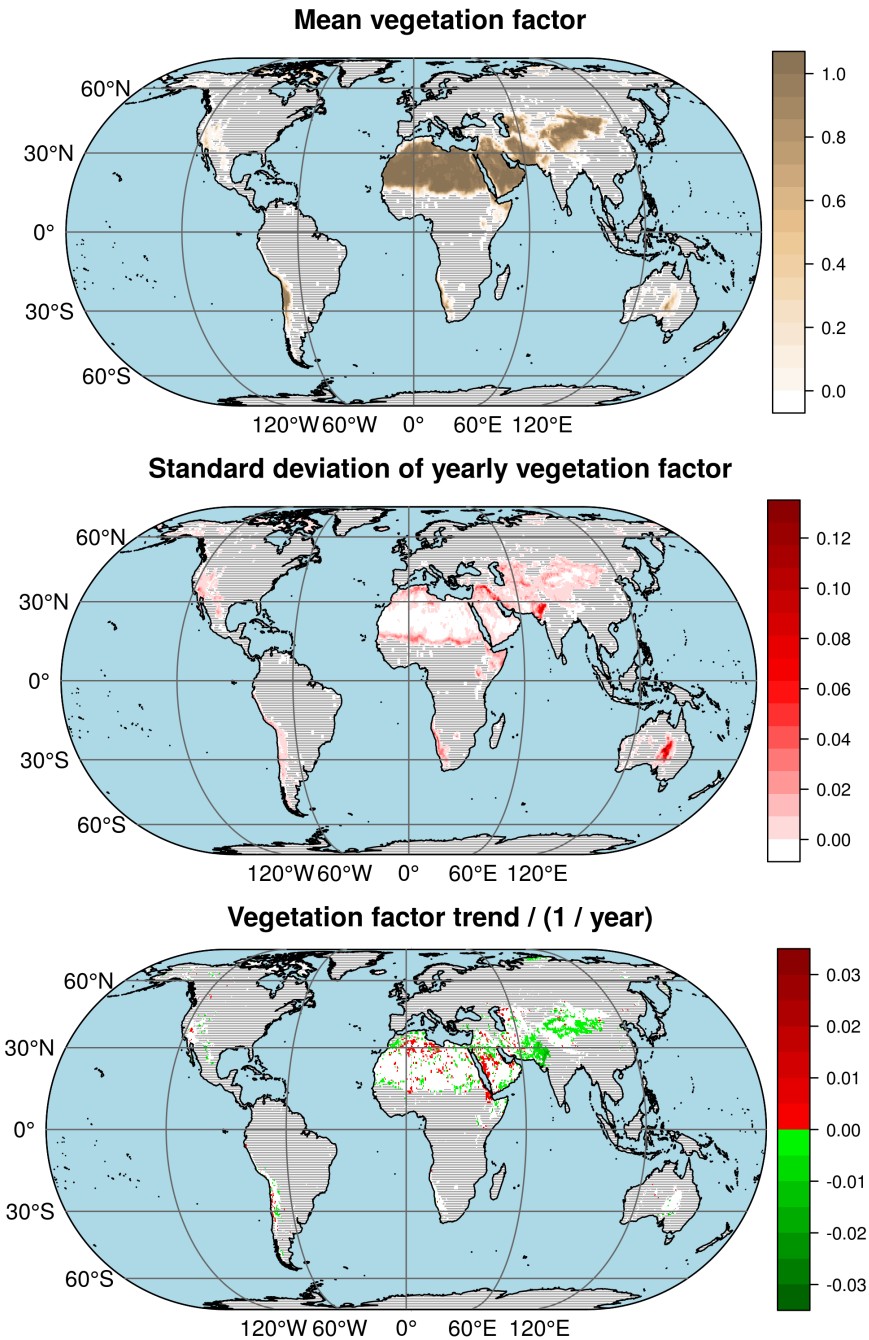

**Figure 2.** Vegetation factor based on leaf area index data from Yuan et al. (2011) averaged over the period 2000 to 2015 (top), the standard deviation of the annual mean values (center) and the trend of the annual mean values (bottom). Regions where the landcover mask precludes emissions throughout the period of available landcover data (2001 to 2012) are hatched.

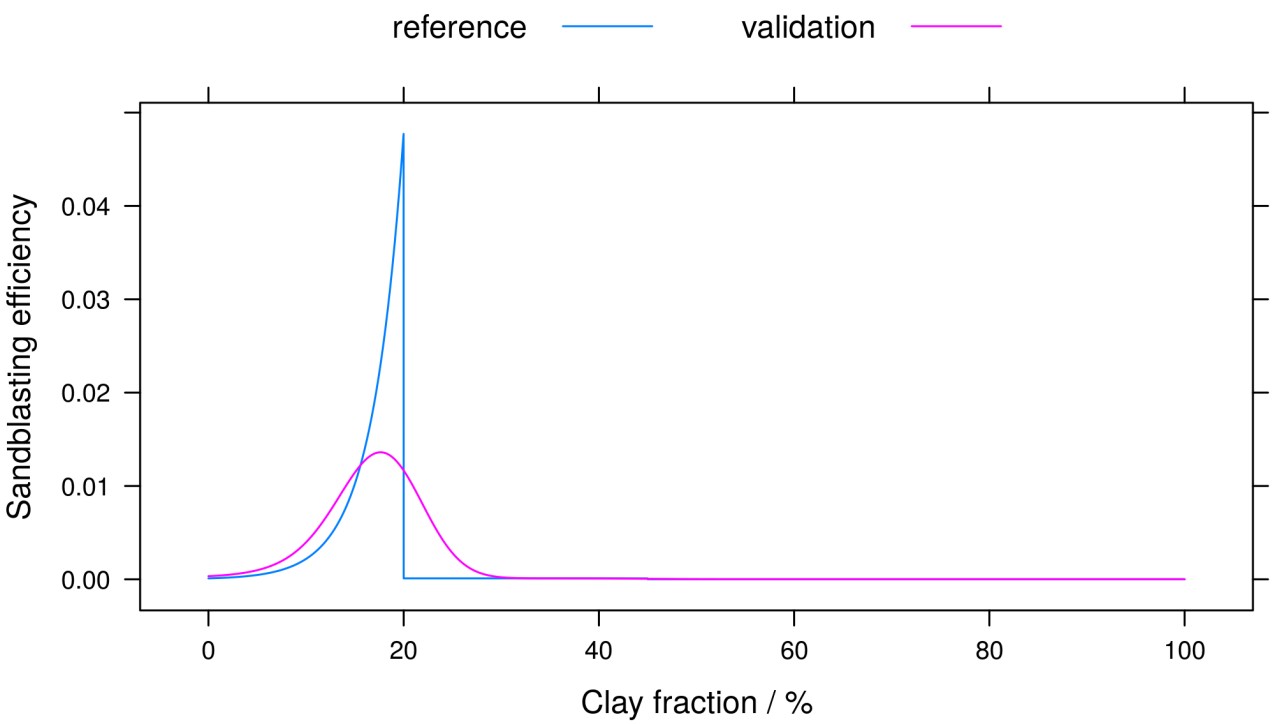

**Figure 3.** The sandblasting efficiency as function of the clay fraction used by Astitha et al. (2012), before ("reference") and after ("valida-tion") applying a Gaussian filter with an interquartile range of 5 %.

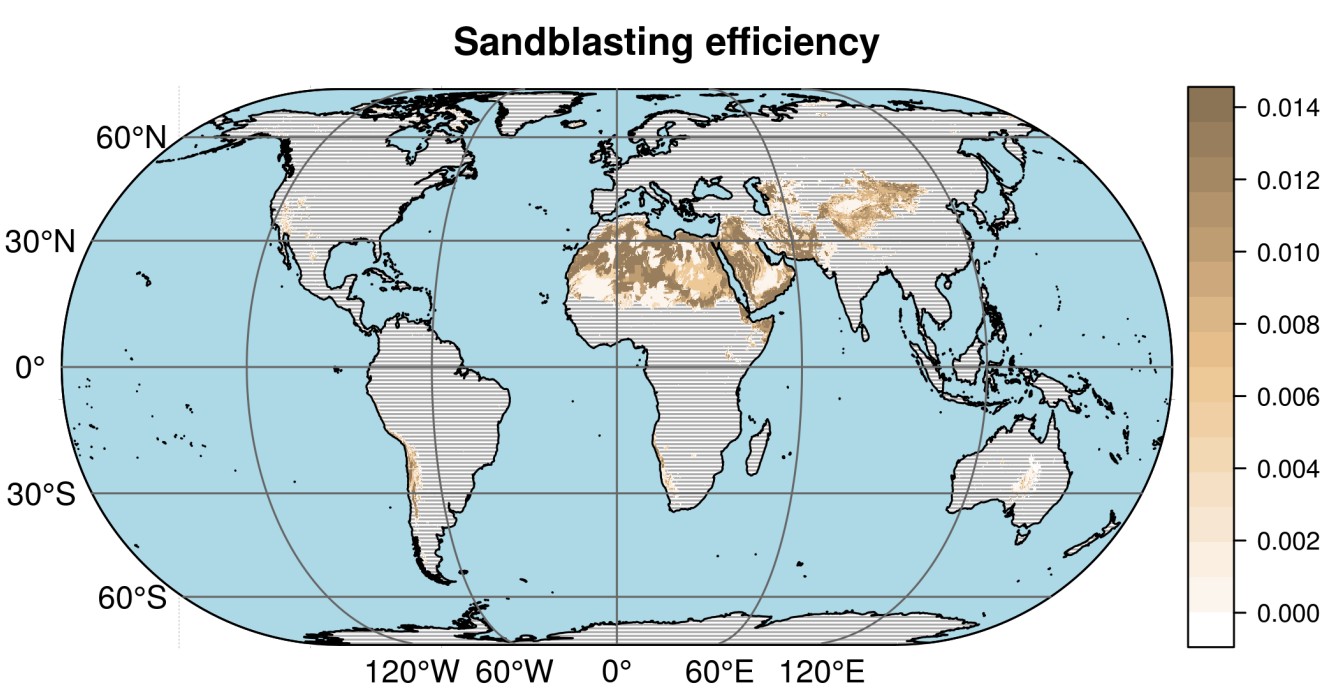

**Figure 4.** Global map of the sandblasting efficiency obtained by applying the filtered efficiency function shown in Fig. 3 to the GSDE clay fraction data. Regions where the landcover mask precludes emissions throughout the period of available landcover data (2001 to 2012) are hatched.

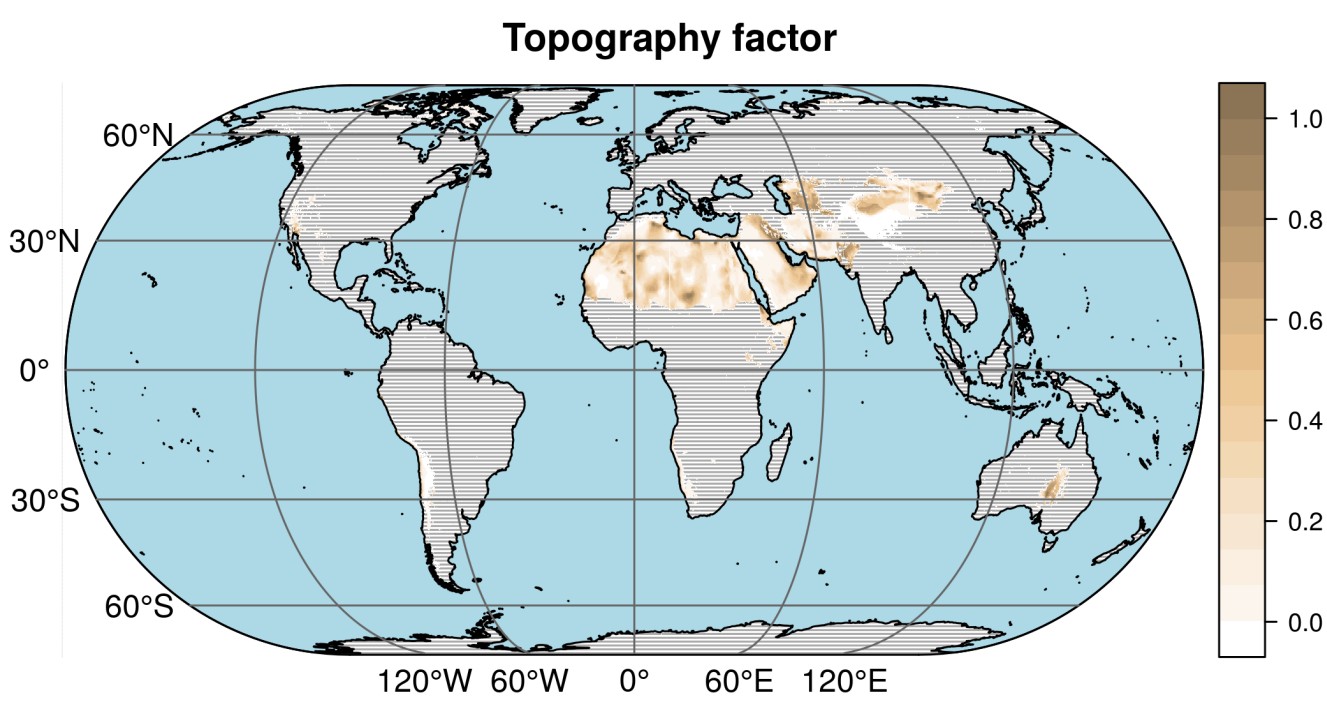

**Figure 5.** The topography factor defined by Eq. 3, calculated using the GMTED2010 elevation data. Regions where the landcover mask precludes emissions throughout the period of available landcover data (2001 to 2012) are hatched.

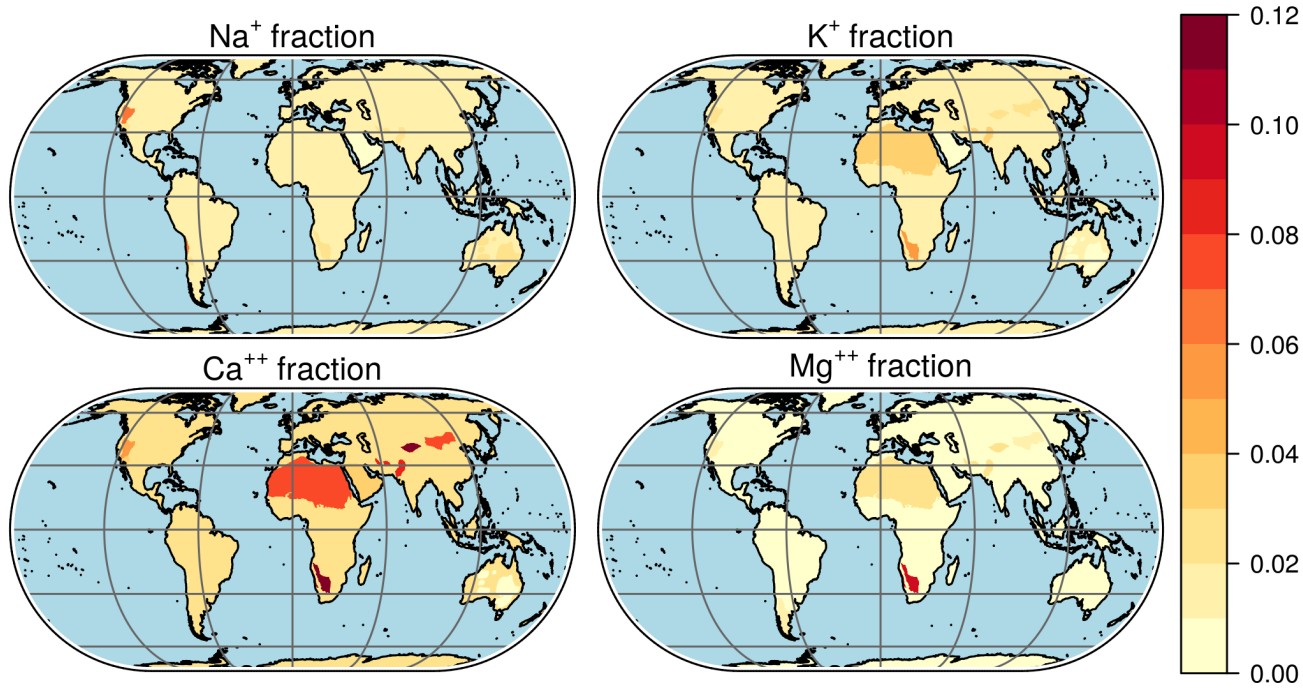

**Figure 6.** Maps of the $Na^+$, $K^+$, $Ca^{++}$ and $Mg^{++}$ mass fractions of the soil of different desert regions, used to calculate the chemical composition of the emitted dust particles.

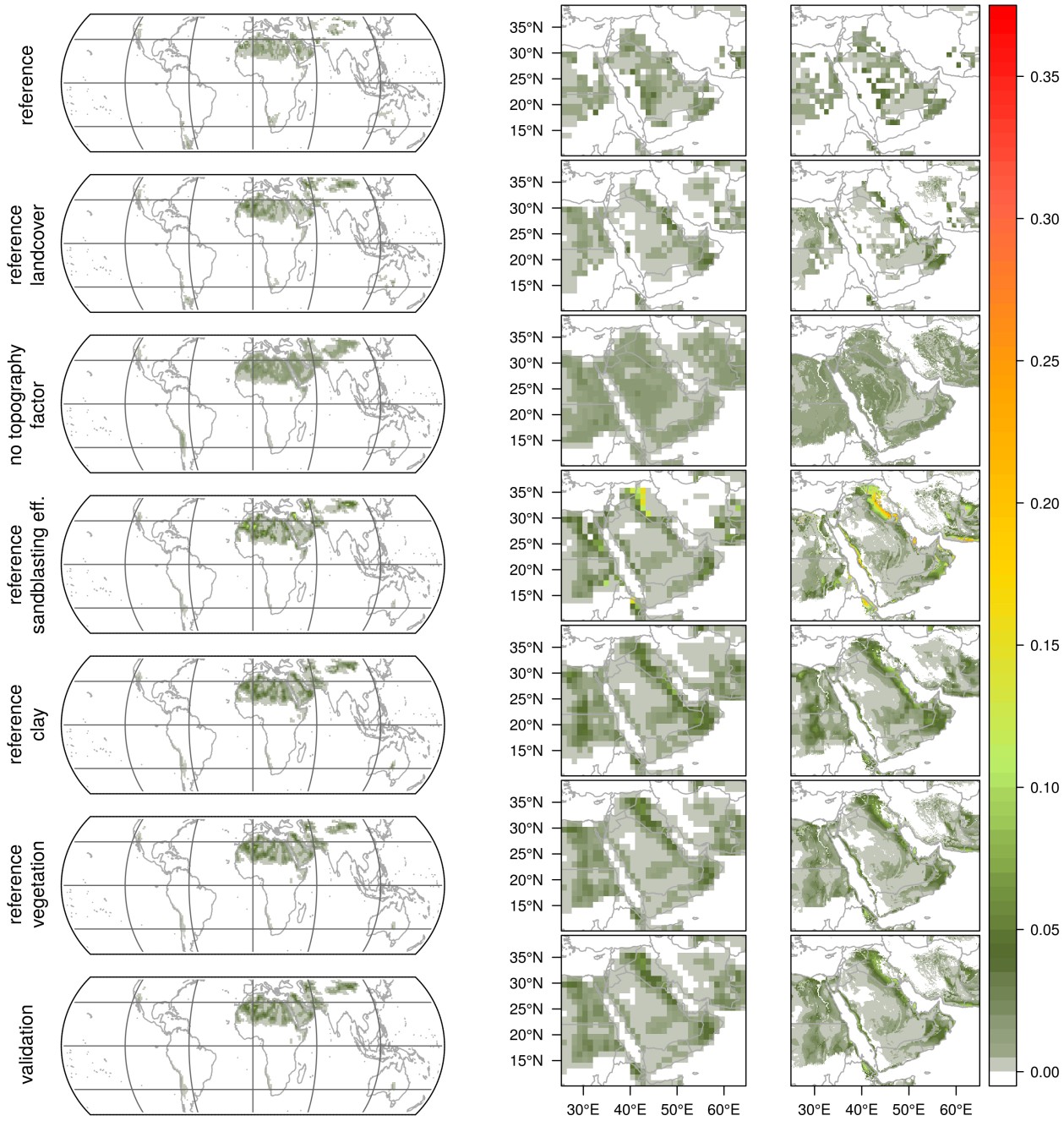

**Figure 7.** Distribution of the emission factor $a$ $f_{\text{landcover}}$ $f_{\text{veg}}$ $N$ $S_{\text{topo}}$ during July 2011 for (from top to bottom) the original emission scheme, the revised emission scheme but using the reference landcover data, no topography factor, the reference sandblasting efficiency, clay fraction or vegetation, and the revised emissions. For the first and second column from the left all data have been regridded to T106 resolution, the third column showing the Middle East illustrates the effect of using the full resolution of the revised input data.

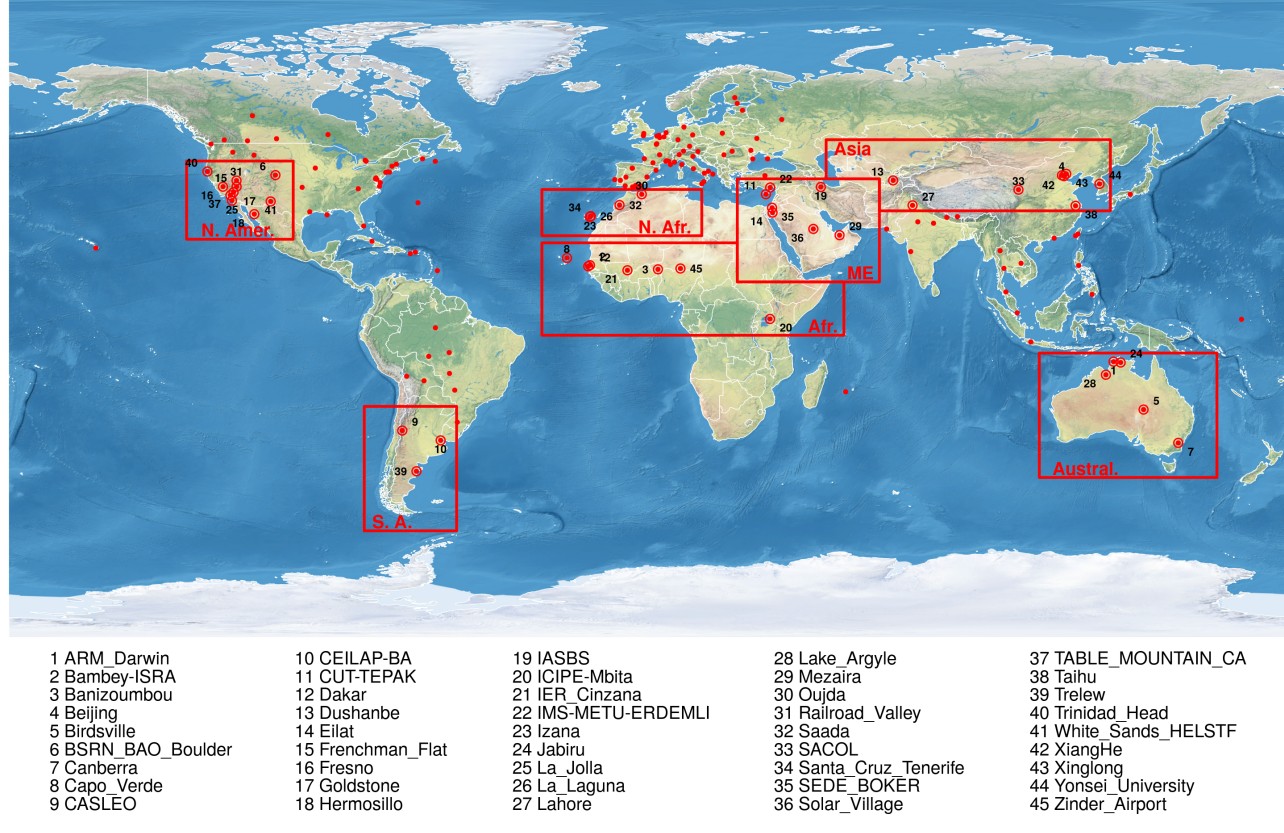

**Figure 8.** AERONET stations and regions of interest used for the evaluation. Stations with data for 120 or more days distributed over at least 9 months of 2011 (red dots) are considered, yielding 45 stations within the regions of interest (labelled).

| | | | | |
|---|---|---|---|---|
| 1 ARM_Darwin | 10 CEILAP-BA | 19 IASBS | 28 Lake_Argyle | 37 TABLE_MOUNTAIN_CA |
| 2 Bambey-ISRA | 11 CUT-TEPAK | 20 ICIPE-Mbita | 29 Mezaira | 38 Taihu |
| 3 Banizoumbou | 12 Dakar | 21 IER_Cinzana | 30 Oujda | 39 Trelew |
| 4 Beijing | 13 Dushanbe | 22 IMS-METU-ERDEMLI | 31 Railroad_Valley | 40 Trinidad_Head |
| 5 Birdsville | 14 Eilat | 23 Izana | 32 Saada | 41 White_Sands_HELSTF |
| 6 BSRN_BAO_Boulder | 15 Frenchman_Flat | 24 Jabiru | 33 SACOL | 42 XiangHe |
| 7 Canberra | 16 Fresno | 25 La_Jolla | 34 Santa_Cruz_Tenerife | 43 Xinglong |
| 8 Capo_Verde | 17 Goldstone | 26 La_Laguna | 35 SEDE_BOKER | 44 Yonsei_University |
| 9 CASLEO | 18 Hermosillo | 27 Lahore | 36 Solar_Village | 45 Zinder_Airport |

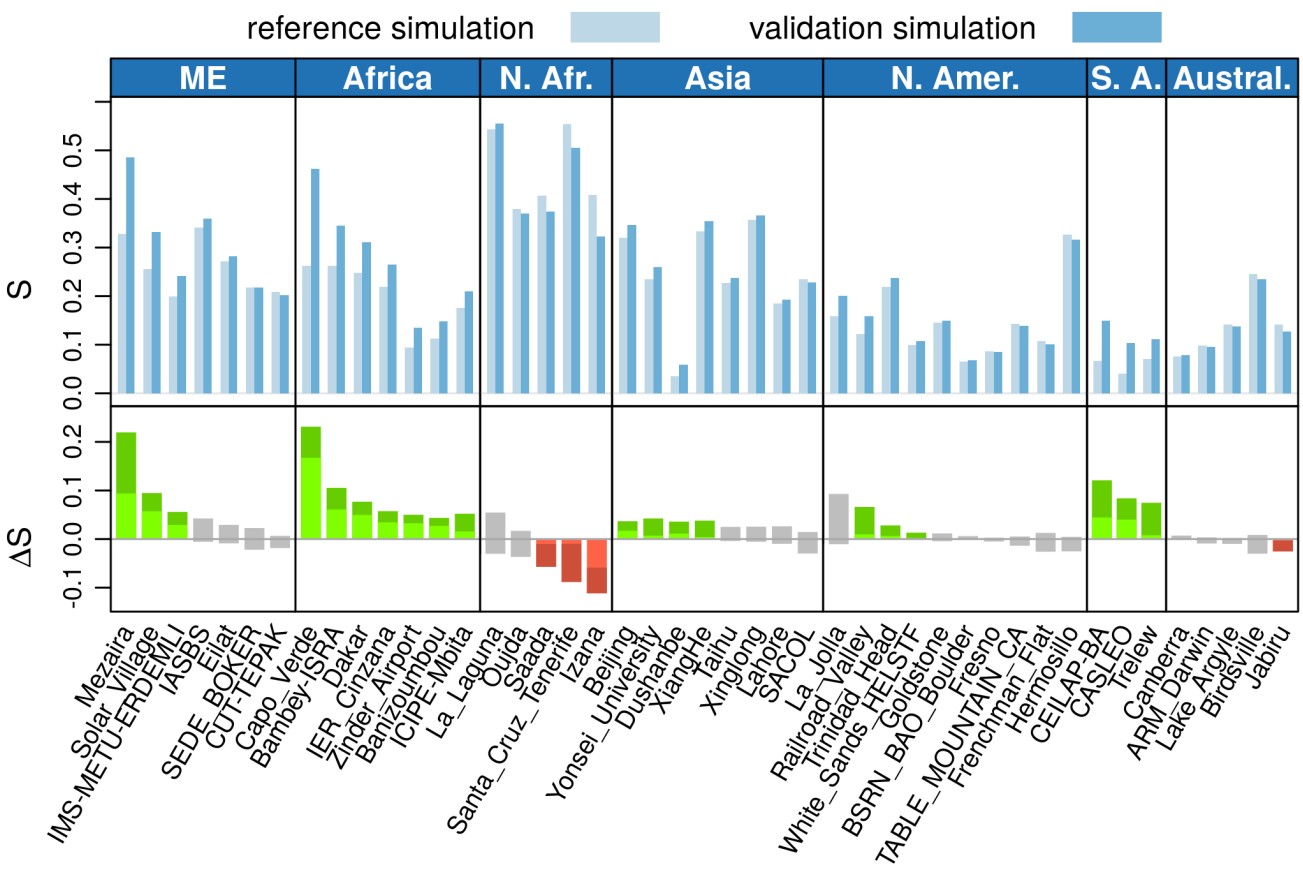

**Figure 9.** Skill score $S$ of the daily mean 550 nm AOD from reference and validation simulations using AERONET observations as benchmark. The red, green and grey bars depict the differences between reference and validation values, with green bars indicating that the validation results agree more closely with the measurements by at least one standard deviation $\sigma$. The corresponding error intervals are indicated by darker colours. Generally, the validation simulation performs better than the reference simulation; regarding the decreased skill scores in north-west Africa, please refer to the discussion in the main text.

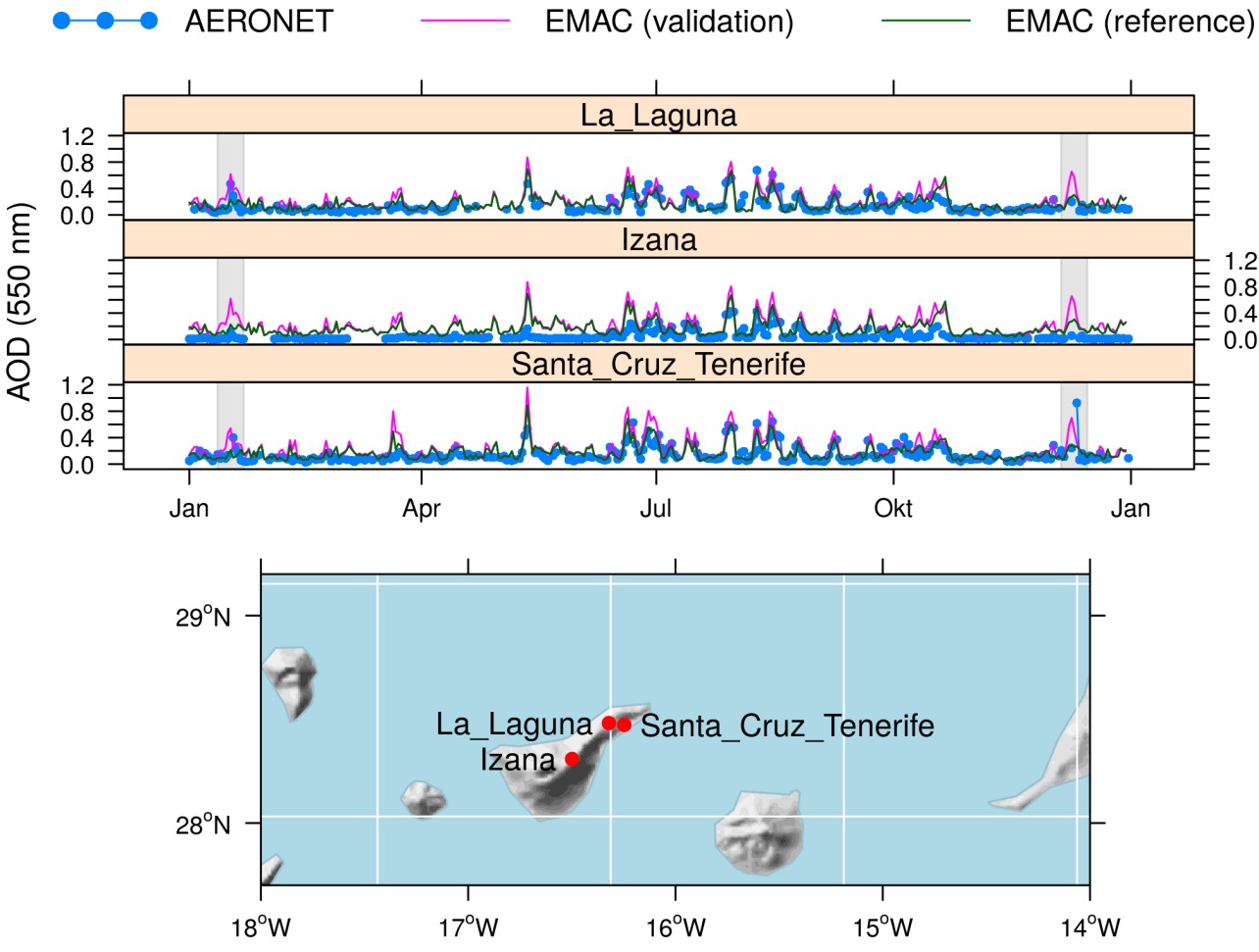

**Figure 10.** Time series of the daily mean AOD at the Canarian AERONET stations (top) and a map showing the location of the stations (bottom). The white squares depict the T106 model grid. During the grey shaded periods of the time series in January and December, at least one of the three AERONET stations observed an AOD peak which is reproduced by the validation but not by the reference simulation.

## 550 nm AOD

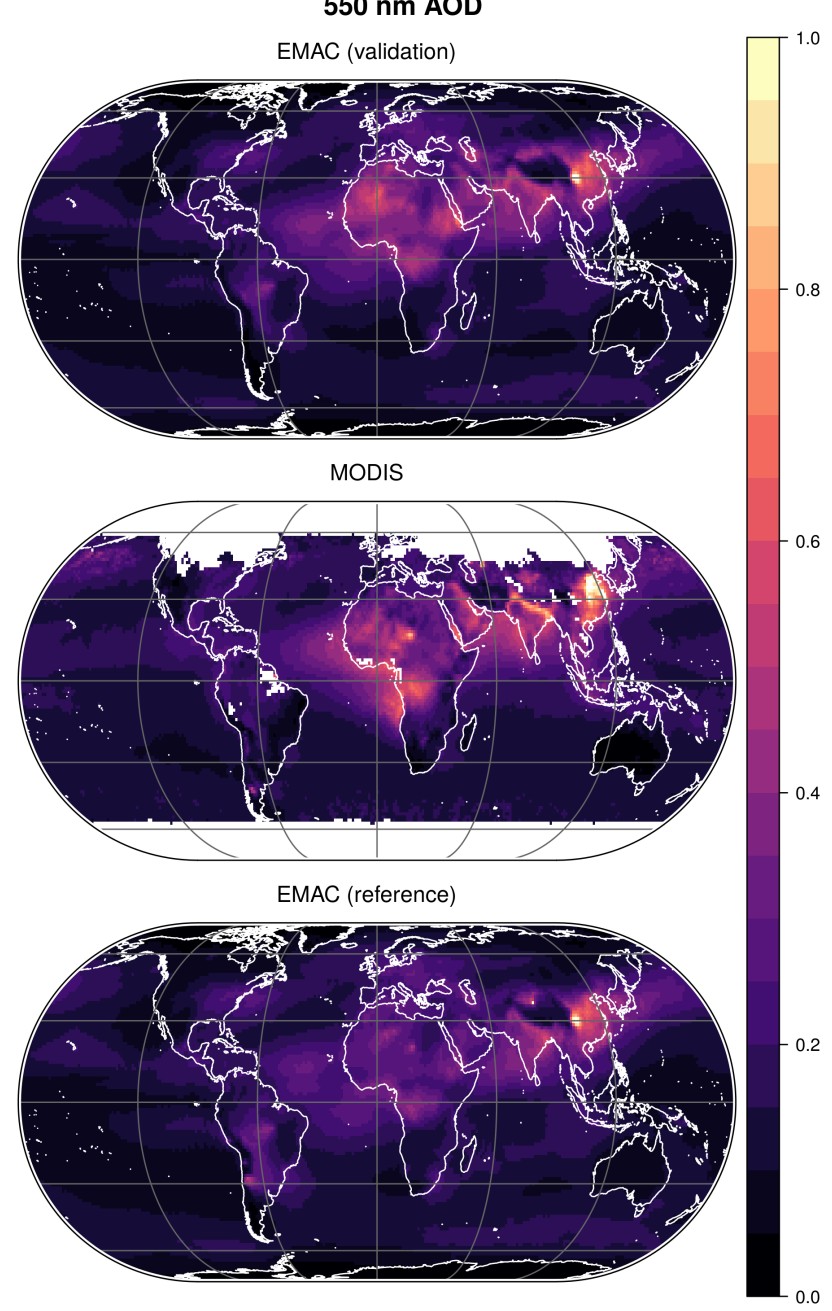

**Figure 11.** Annual mean for 2011 of the AOD at 550 nm wavelength observed by MODIS (centre) and simulated by EMAC with ("valida­tion", top) and without ("reference", bottom) revision of the dust emission scheme. The revised dust emissions enhance the correlation of the AOD pattern from 0.79 to 0.81, the skill score from 0.58 to 0.67.

# 550 nm AOD

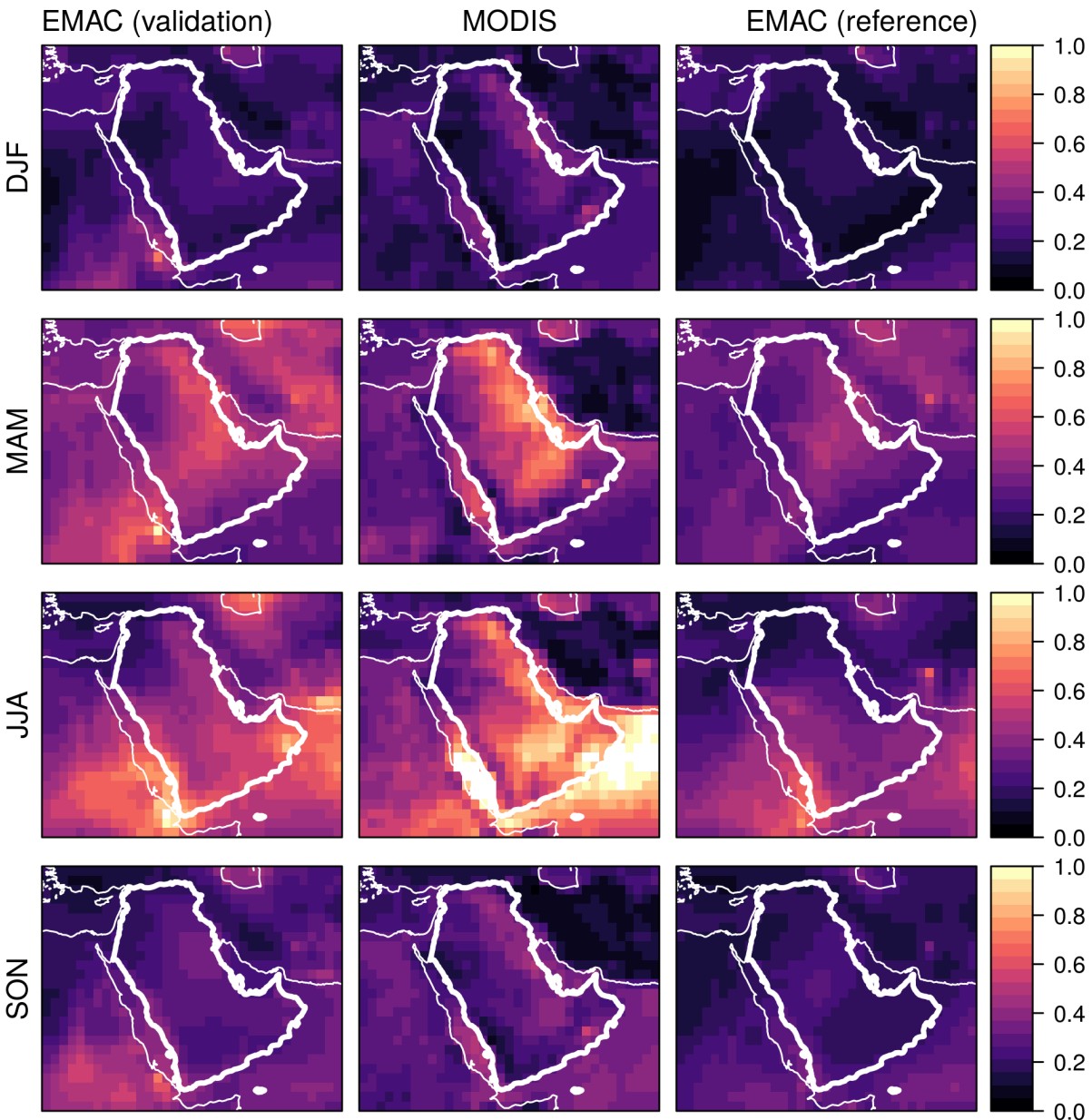

**Figure 12.** Seasonal 550 nm AOD over the Middle East (region of interest A) in 2011 observed by MODIS (centre column) and simulated by EMAC with ("validation", left) and without ("reference", right) revision of the dust emission scheme. Each row shows the three-month averages over the periods (from top to bottom) DJF (December, January, February), MAM (March, April, May), JJA (June, July, August) and SON (September, October, November). Especially throughout the white-bounded region encompassing the Arabian Peninsula including Iraq, Syria and Jordan the AOD distribution obtained with the revised dust emissions agrees significantly better with the MODIS observations (see Fig. S6 in the supplement).

# 10 um DAOD

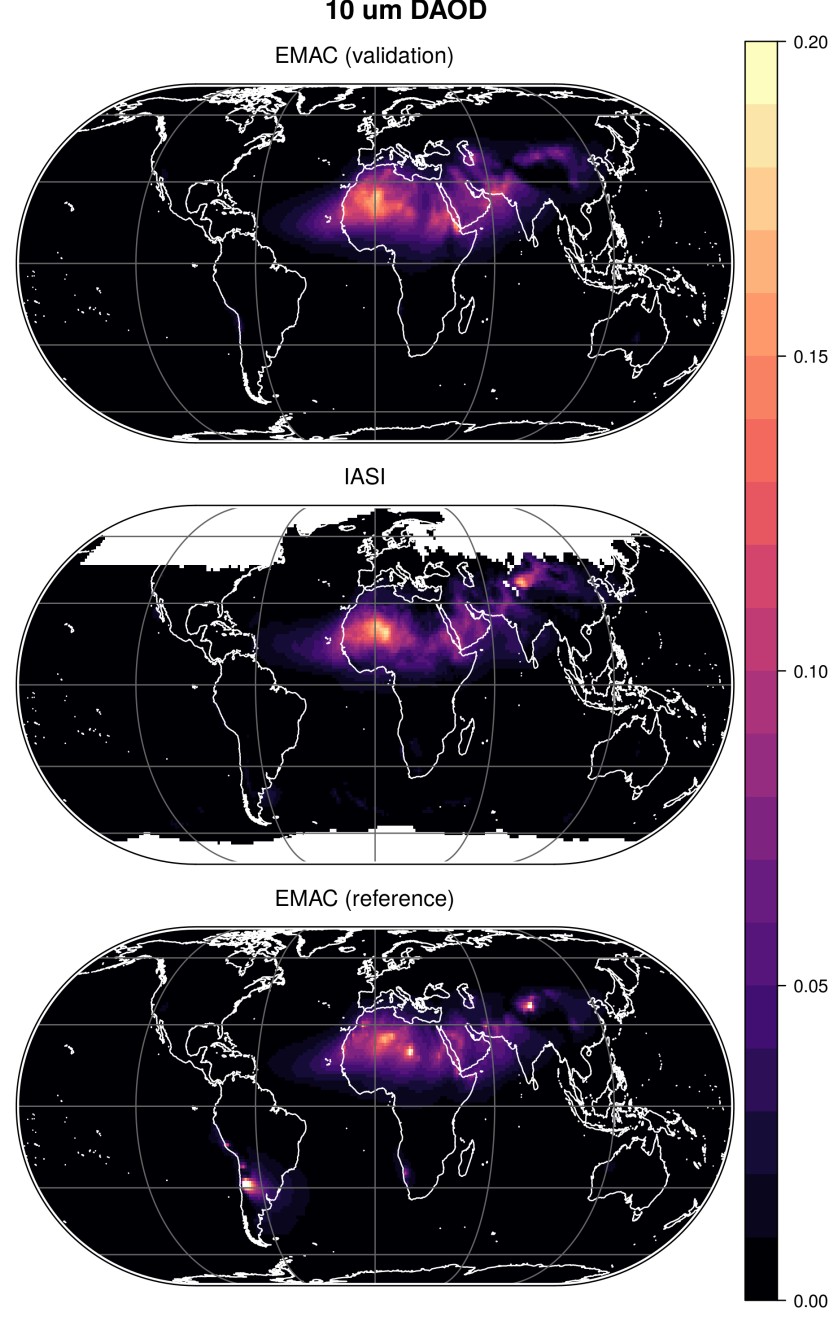

**Figure 13.** Annual mean for 2011 of the DAOD at 10 $\mu$m wavelength observed by IASI (centre) and simulated by EMAC with ("validation", top) and without ("reference", bottom) revision of the dust emission scheme. The revised dust emissions enhance the correlation of the AOD pattern from 0.79 to 0.89, the skill score from 0.64 to 0.78.

# 10000 nm DAOD

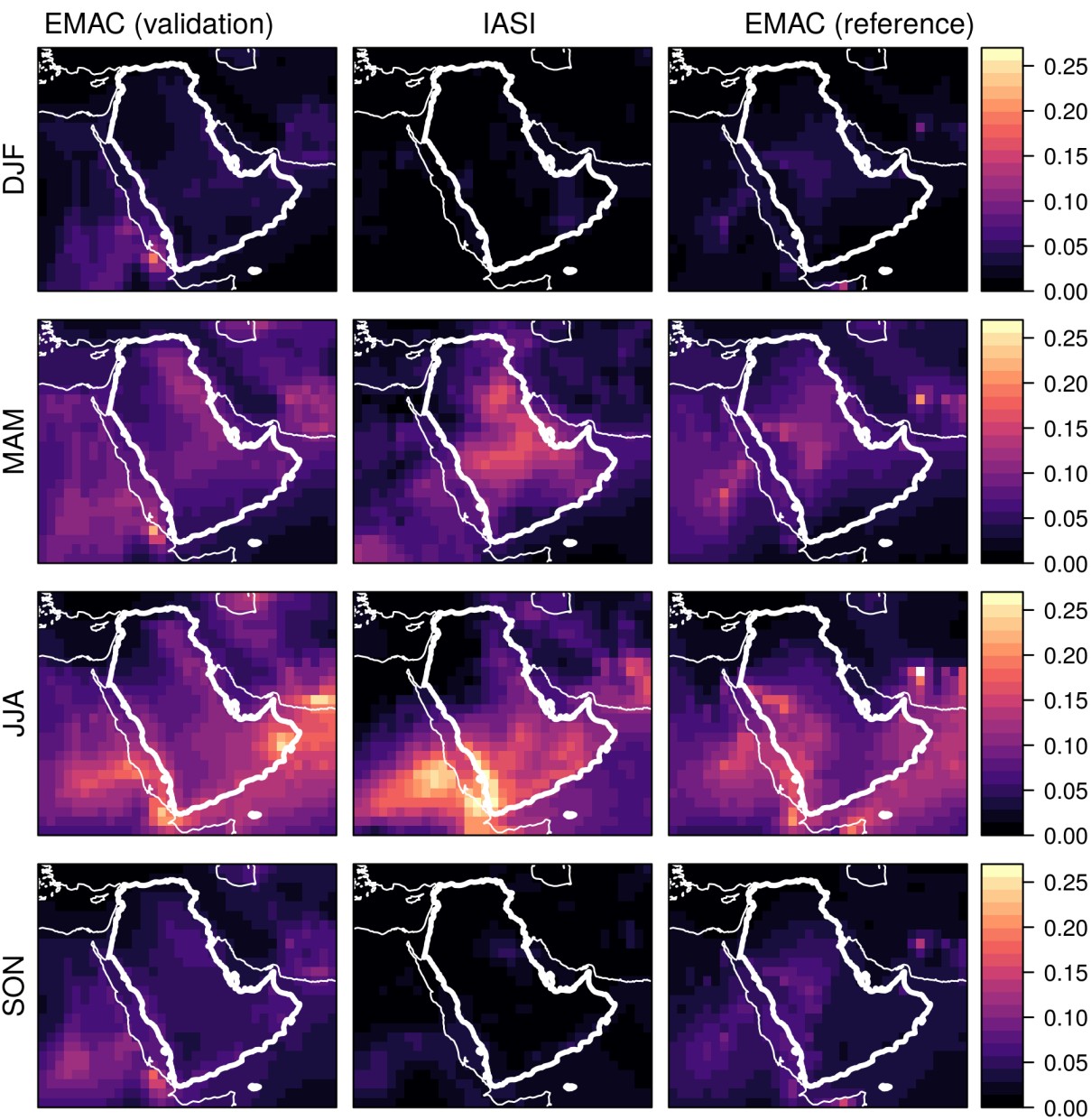

**Figure 14.** Seasonal 10 $\mu$m DAOD over the Middle East (region of interest A) in 2011 observed by IASI (centre column) and simulated by EMAC with ("validation", left) and without ("reference", right) revision of the dust emission scheme. Each row shows the three-month averages over the periods (from top to bottom) DJF (December, January, February), MAM (March, April, May), JJA (June, July, August) and SON (September, October, November). Especially throughout the white-bounded region encompassing the Arabian Peninsula including Iraq, Syria and Jordan the DAOD distribution obtained with the revised dust emissions agrees significantly better with the IASI observations (see Fig. S7 in the supplement).

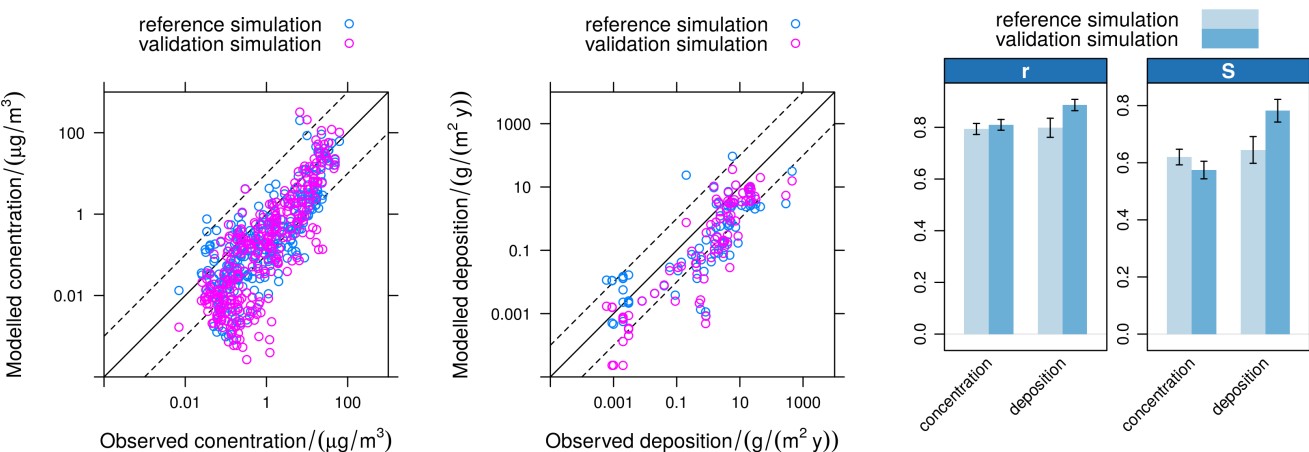

**Figure 15.** Comparison of modelled and observed dust concentration and deposition: scatterplots of monthly concentrations (left) and annual deposition (centre), and barcharts of the corresponding correlation coefficients $r$ and skill scores $S$ (right). The observations are taken from the AEROCOM dust benchmark (Huneeus et al., 2011).

**Table 1.** Summary of updated and added input data

|  |  | Reference input data | Updated/new input data |
|---|---|---|---|
| Land cover | Source | Olson (1992) | MODIS MCD12C1 |
|  | Spatial resolution | 1°(aggregated from 10') | 0.05° |
|  | Temporal resolution | static | yearly data (since 2001) |
| Clay fraction | Source | Scholes and Brown de Colstoun (2011) | GSDE (Shangguan et al., 2014) |
|  | Spatial resolution | 1° | 0.1°(aggregated from 30") |
|  | Temporal resolution | static | static |
|  | Notes | clay fraction in top 30 cm soil layer | clay fraction in top 4.5 cm soil layer |
| Vegetation | Source | Kergoat et al. (1999); Bonan et al. (2002) | Yuan et al. (2011) |
|  | Spatial resolution | 1°(aggregated from 0.5°) | 0.1°(aggregated from 30") |
|  | Temporal resolution | monthly values (Apr 1992 to Mar 1993) | monthly values (since 2000, aggregated from 8 day values) |
|  | Notes |  | MODIS based |
| Topography | Source | - | Danielson and Gesch (2011); GMTED2010 (2010) |
|  | Spatial resolution | - | 0.1°(aggregated from 30") |
|  | Temporal resolution | - | static |
| Chemical composition | Source | - | Karydis et al. (2016); Natural Earth (2016) |
|  | Spatial resolution | - | 0.1° |
|  | Temporal resolution | - | static |

**Table 2.** Parameters of emission and GMXE dust modes. The GMXE parameter values shown have been used for reference and validation simulation.

| | $\sigma_g$ | $\tilde{d}/\mu\mathrm{m}$ | $d_{\mathrm{min}}/\mu\mathrm{m}$ | $d_{\mathrm{max}}/\mu\mathrm{m}$ |
|---|---|---|---|---|
| Emission modes | 2.1 | 0.83 | | |
| | 1.9 | 4.82 | | |
| | 1.6 | 19.4 | | |
| GMXE dust modes | 1.59 | | 0.12 | 2 |
| | 2 | | 2 | $\infty$ |

**Table 3.** Global mineral dust emissions in 2011 obtained by EMAC.

|                    | Validation simulation | Reference simulation |
| ------------------ | --------------------- | -------------------- |
| Accumulation mode  | 0.148 Gt / year       | 0.0517 Gt / year     |
| Coarse mode        | 1.16 Gt / year        | 1.28 Gt / year       |
| Total              | 1.31 Gt / year        | 1.33 Gt / year       |