# Peer review of "Revised mineral dust emissions in the atmospheric"

_Geoscientific Model Development, 2017_

## Short Comment (SC1) · 18 Jul 2017

Dear authors,

In my role as Executive editor of GMD, I would like to bring to your attention our Editorial version 1.1:

http://www.geosci-model-dev.net/8/3487/2015/gmd-8-3487-2015.html

This highlights some requirements of papers published in GMD, which is also available on the GMD website in the 'Manuscript Types' section:

http://www.geoscientific-model-development.net/submission/manuscript_types.html

In particular, please note that for your paper, the following requirements have not been met in the Discussions paper:

- "The main paper must give the model name and version number (or other unique identifier) in the title."

- "All papers must include a section, at the end of the paper, entitled 'Code availability'. Here, either instructions for obtaining the code, or the reasons why the code is not available should be clearly stated. It is preferred for the code to be uploaded as a supplement or to be made available at a data repository with an associated DOI (digital object identifier) for the exact model version described in the paper. Alternatively, for established models, there may be an existing means of accessing the code through a particular system. In this case, there must exist a means of permanently accessing the precise model version described in the paper. In some cases, authors may prefer to put models on their own website, or to act as a point of contact for obtaining the code. Given the impermanence of websites and email addresses, this is not encouraged, and authors should consider improving the availability with a more permanent arrangement. After the paper is accepted the model archive should be updated to include a link to the GMD paper."

Consequently, I ask you to take the following points into account:

- In the title: "based on MESSy_2.52": please identify the exact version you are describing in your article. As you updated the dust emission scheme it would be good to have a label for the dust emission scheme in the title: e.g. "The revised mineral dust emission scheme DU_Asthita (v2.0) in the atmospheric chemistry-climate model EMAC an advancement of MESSy_2.52". Additionally, please note, that this exact Code version must be permanently archived.

- Code and Data availability sections: In GMD the code and data described in the article has to be made publicly available as long as no licence reasons prevent this. Therefore, please upload the data and the code to a public repository providing a DOI for the data or state explicitly, why the data / code can not be made freely available. (In your case the MESSy homepage provides a template for the code availability section in GMD. Why are you not using it?) Please also state if the update will become a part of the official MESSy code version.

Yours,

Astrid Kerkweg

---

## Referee Comment (RC1) · Anonymous Referee #1 · 1 Sep 2017

This paper entitled "Revised mineral dust emissions in the atmospheric chemistry-climate model EMAC (based on MESSy 2.52)" and submitted to GMD presents new developments concerning the parameterization of dust emissions in the global model ECHAM/MESSy. These new developments have been evaluated and compared to the previous version of the model in terms of the resulting aerosol optical depth. The use of ground-based (AERONET) and satellite (MODIS, IASI) has shown the improvement brought by this new version. This paper is therefore interesting for the community working on dust modeling, and the manuscript is well written. However, the current version needs major revision before considering the publication in GMD because of the following points:

[Figure]

- The evaluation of the revised emissions is limited to the aerosol optical depth, which is not enough to estimate the quality of the parameterization. AOD is indeed a relevant parameter to evaluate the integrated effect of dust aerosols on radiation, but it can hinder some compensating errors. Besides, such parameters as the dust size distribution, the dust vertical profile or dust deposition are essential for radiative budget and effects on climate, and are not constrained by AOD. The authors could for example add an evaluation of surface concentrations, dust deposition, dust emission fluxes or dust vertical profiles, as done by similar recent studies (Kok et al., 2014; Albani et al., 2014; Klose et al., 2014; Gherboudj et al., 2015).

- As there are many papers on dust modeling, the authors should highlight more the originality of their work. In this purpose, they should add a paragraph in the introduction presenting the state-of-the-art in dust modeling in global chemistry-climate models. This would be useful for the whole community, and would not restrain the impact of the paper to the ECHAM community as it could be the case with the current version of the paper.

Specific comments:

- Abstract: The authors mention several times the possibility to run high resolution simulations. What is the targeted resolution? Do the scheme need any modification for this high resolution?

- Page 2 Lines 18-19: The authors should justify the "rapid changes of deserts and semi-arid regions in recent decades"

- Page 3 Section 2.1: Looking at Fig.1, I get the impression that there are more regions with shrinking deserts, is it true?

- Page 3 Line 27: Any justification for the equation (1) giving the vegetation factor? Is it used in other models?

- Page 3 Lines 29-30: Could the authors clarify which statistical test they have used?

- Page 4 Section 2.3: Contrary to Sections 2.1 and 2.2, the authors have not elaborated on the differences between the two versions of the clay fraction maps. Which is the expected impact on dust emissions?

- Page 4 Lines 26-29: Is there any work forecast to include again the effect of soil moisture on dust emissions? It might be important in some regions like Sahel.

- Page 5 Line 5: This equation differs from the one given in Astitha et al. (2012), the authors should correct it or explain why it is different.

- Page 5 Line 9: The authors should justify the choice of 0.4 m/s, and clarify what they call "good results" explaining what has been compared.

- Page 5 Line 29: The parameter dmax could be added in Table 2.

- Page 6 Lines 10-15: I did not understand if finally the chemical composition of dust is included or not in the model.

- Page 6 Lines 19-21: The list of submodels is unclear for readers not familiar to the model. The authors should add a reference to have the details about these parameterizations.

- Page 6 Line 25: What is the Tanré climatology used for? (AOD or only other optical properties?)

- Page 6 Line 30: A reference to Table 1 should be added to present the simulations.

- Page 6 Line 32: Is a one-year simulation long enough to evaluate the revised dust emissions? Is there any reason to select the year 2011?

- Page 7 Line 15: Which level of AERONET AOD has been used in this comparison?

- Page 7 Line 17: Maybe the authors should divide the region B in two sub-regions, for the reader to identify more easily the different stations.

- Page 7 Lines 29-30: I don't understand how this skill score based on correlation can

be affected by a bias.

- Page 7 Section 4.1: It could be also useful to add one or two time series in stations where the score has increased.

- Page 8 Line 1: Which is the altitude of the model grid cell?

- Page 8 Line 21: Is this increase of spatial correlation statistically significant?

- Page 8 Lines 28-30: The authors could think about adding a score for the measuring the improvement in seasonal cycle, which could reinforce the robustness of their results.

- Page 9 Line 17: Same remark for the significance of the increase in the skill score.

- Page 9 Line 32: The time dependence of land cover and vegetation has not been tested here because the simulations were too short.

Technical comments:

- Page 2 Lines 8-9: The abbreviations DU_Astitha1 and DU_Astitha2 are useless since they are not used in the rest of the paper.

- Page 6 Line 24: ISORROPIA

- Figure 1: The color bar should be changed, because the values below -0.2 cannot be distinguished.

- Figure 8: The authors could replace the letters (A, B, etc.) by the name of the regions in the blue line at the top of the figure.

- Figure 9: AERONET data is represented with dots in the figure, while it is a line in the caption.

- References: The format needs to be homogenized (notably the use of first names for the first author).

References:

Albani, S., N. M. Mahowald, A. T. Perry, R. A. Scanza, C. S. Zender, N. G. Heavens, V. Maggi, J. F. Kok, and B. L. Otto-Bliesner (2014), Improved dust representation in the Community Atmosphere Model, J. Adv. Model. Earth Syst., 6, 541–570, doi:10.1002/2013MS000279.

Astitha, M., J. Lelieveld, M. Abdel Kader, A. Pozzer, and A. de Meij (2012), Parameterization of dust emissions in the global atmospheric chemistry-climate model EMAC: impact of nudging and soil properties. Atmospheric Chemistry and Physics, 12(22):11057–11083, doi:10.5194/acp-12-11057-2012.

Gherboudj, I., S. N. Beegum, B. Marticorena, and H. Ghedira (2015), Dust emission parameterization scheme over the MENA region: Sensitivity analysis to soil moisture and soil texture, J. Geophys. Res. Atmos., 120, 10,915–10,938, doi:10.1002/2015JD023338.

Klose, M., Y. Shao, X. Li, H. Zhang, M. Ishizuka, M. Mikami, and J. F. Leys (2014), Further development of a parameterization for convective turbulent dust emission and evaluation based on field observations, J. Geophys. Res. Atmos., 119, 10,441–10,457, doi:10.1002/2014JD021688.

Kok, J. F., Mahowald, N. M., Fratini, G., Gillies, J. A., Ishizuka, M., Leys, J. F., ... & Zobeck, T. M. (2014). An improved dust emission model–Part 1: Model description and comparison against measurements. Atmospheric Chemistry and Physics, 14(23), 13023-13041.

---

## Referee Comment (RC2) · Anonymous Referee #2 · 11 Sep 2017

General comments

This study presents updates of the dust emission scheme implemented in the global atmospheric chemistry model EMAC based on the previous work of Astitha et al. (2012). The land cover, vegetation topography and clay fraction maps are updated to more recent versions using higher spatial resolution. Changes are also imposed to the dust emission scheme directly. The updated dust emissions are evaluated with AOD measurements from AERONET, MODIS and IASI for the year 2011. The title, flow and structure of the paper are appropriate. All updates are well received and long needed, given the importance of quality input data to accurately parameterize physical processes that cannot be described by first principles. However, the authors keep the evaluation part largely on the qualitative side, which does not help the reader and the community to fully understand why these changes were impactful and significant. The conclusions are also very brief for a model development/improvement paper. The authors miss a great opportunity to discuss the very interesting aspects of each revision and inform the community of which one should be considered more impactful (if not all). The specific comments below will help the authors revise the paper so it can be accepted for publication with GMD.

Specific comments/suggestions

Section 2.2 (Vegetation): Please elaborate on the calculation of fveg (Eq.1): What is the role of 0.35 and what is the meaning of fveg being 1 or less than 1.

Section 2.3 (Clay fraction): provide a map of the updated clay fraction in comparison to the one previously used in the model. It will provide context on the significance of changes that later affect the parameterization scheme.

Section 3 Page 4 (soil moisture): The soil moisture term in Astitha et al. (2012) and Eq. A1 in this paper is omitted from the threshold friction velocity. However, the authors correctly describe the dependence of dust emission on soil moisture at the end of this paragraph. What is not clear is if the statement "we consider a detailed parametrisation of the soil moisture effect to be essential to capture the observed trends in future simulations. This will require a comprehensive soil model providing accurate moisture values for the topmost surface layer" refers to an action already taken for this study or a future goal. In any case, a discussion on how the exclusion of soil moisture correction influences the simulations is important here. Page 5, (Surface friction velocity limit): a note must be placed that Eq.2 holds only when $u^* > u^*_t$. Also, choosing to limit the threshold velocity to a maximum value of 0.4 m/s seems arbitrary and needs to be elaborated. What led the authors to this specific value? Some context and rationale must be provided. Page 5 (Topography factor): I am not sure of the role of the normalization factor 5.3 and how it conserves global emissions. It sounds like a tuning factor to me, so please elaborate on the role of the factor and the method used to estimate it. Page 5 (Mode mapping): This is not a strict update of the emission scheme but rather an alteration in order to use the GMXE aerosol model compared to M7 used by Astitha et al. (2012). A brief note must be included in this section to clarify that the original scheme (as well as the reference simulation herein) used a different aerosol module thus a different approach to particle size distribution. The omission of the eight transport size bins is surely a change from the original version. Figure 6: what is the higher value in this scale (above 0.1)? When we see 0.1 fraction of Ca++, does that indicate the mass, volume (or else) fraction of the total particles within each specific grid cell? A better explanation of the mineral cations fraction could be included also in page 6 (last paragraph of section 3).

Section 4.1, page 7: 1. "On the other hand, dust events observed by AERONET in January and December are reproduced by the validation simulation, but not by the reference simulation": this comparison is not at all discernible in the plot as it is. If this is an important argument, the plot must be revised somehow to make the statement visible. 2. Given that Izana and La Laguna are within the same model grid cell, an average of the AOD from both sites could be an alternative way to compare with the model value. In addition, when evaluating numerical model simulations one can employ the nearest neighbor (as done here) or a bilinear interpolation between the observation and the model value from the four closest grid points. 3. The main criticism I have for the evaluation using the skill score (as with any other statistical metric) is the qualitative determination of which configuration provided the best results. Characterizations such as "slightly better" or "marginally larger" do not show robustness in the performed evaluation. My immediate question is: are these differences statistically significant? Are they statistically different? This is the only way to prove or convince the audience that a, say, 0.05 change in the skill score is significant enough. 4. What about using AOD of coarse vs. accumulation modes from AERONET? 5. How about using total PM concentrations wherever available (and for cases of high dust concentrations) to

evaluate model performance? An additional means of quantitative evaluation needs to be included.

Figure 8: Please consider replacing "Regions A, B," etc. from the figure with the names of the regions as it is not convenient to go back and forth between Fig. 7 and 8 to identify the regions.

Sections 4.2 and 4.3: The scarcity of desert dust concentration measurements is a well-known problem in the modeling community when assessing model and parameterization scheme performances. This is when satellite and remote sensing observations come into play and are important tools of assessment. Nevertheless, leaving the comparison in the qualitative state only, influences the robustness of the conclusions. Looking at the IASI zoomed plots (Fig. 13), I would not immediately say that the validation is better than the reference simulation. They are different and somewhat both incorrect in my view. If the authors presented a quantitative assessment of the performance, there would be no doubt on the comparison. Also, why is the zoomed area over Middle East only? What is the special interest for this specific region? This has to be explained thoroughly so it will not be seen as "cherry picking".

Conclusions: the conclusions are quite brief (two sentences in the end of the section). More discussion should be invoked on how the changes influence a better model performance (as long as there is a robust determination of "better" or "worse"). There are very interesting aspects in this study and it would be very useful for the community to understand how the changes that were implemented individually affect dust emissions. I believe that adding such discussion would greatly strengthen the paper.

Appendix: There are a couple of things missing in the depiction of the emission flux jemis (Eq A2): 1) I don't see the mass fraction (source to transport bins, M in Eq.4) that should be multiplied in the right side of the equation. 2) In Astitha et al. (2012) they used the relative surface area covered from particles with diameter D (Srel) to calculate the horizontal flux H. Did the authors omit this calculation in their revisions?

---

## Author Comment (AC1) · 17 Nov 2017

**Reply to referee 1**

We thank the referee for the comprehensive and constructive review. All aspects have been addressed in a revised manuscript and a new supplement. Below please find our point-by-point reply to the comments.

> This paper entitled "Revised mineral dust emissions in the atmospheric chemistry-climate model EMAC (based on MESSy 2.52)" and submitted to GMD presents new developments concerning the parameterization of dust emissions in the global model ECHAM/MESSy. These new developments have been evaluated and compared to the previous version of the model in terms of the resulting aerosol optical depth. The use of ground-based (AERONET) and satellite (MODIS, IASI) has shown the improvement brought by this new version. This paper is therefore interesting for the community working on dust modeling, and the manuscript is well written. However, the current version needs major revision before considering the publication in GMD because of the following points:
> - The evaluation of the revised emissions is limited to the aerosol optical depth, which is not enough to estimate the quality of the parameterization. AOD is indeed a relevant parameter to evaluate the integrated effect of dust aerosols on radiation, but it can hinder some compensating errors. Besides, such parameters as the dust size distribution, the dust vertical profile or dust deposition are essential for radiative budget and effects on climate, and are not constrained by AOD. The authors could for example add an evaluation of surface concentrations, dust deposition, dust emission fluxes or dust vertical profiles, as done by similar recent studies (Kok et al., 2014; Albani et al., 2014; Klose et al., 2014; Gherboudj et al., 2015).

We appreciate the advice and have added the evaluation of surface concentrations and dust deposition for an even more comprehensive validation. Regarding the current evaluation we would like to point out that our comparison with the satellite retrieved 10 um DAOD goes beyond the evaluation used in other studies and, combined with the AOD comparison at visible wavelengths, amongst others probes aspects of the particle size distribution.

> - As there are many papers on dust modeling, the authors should highlight more the originality of their work. In this purpose, they should add a paragraph in the introduction presenting the state-of-the-art in dust modeling in global chemistry-climate models. This would be useful for the whole community, and would not restrain the impact of the paper to the ECHAM community as it could be the case with the current version of the paper.

We have extended the introduction accordingly.

> Specific comments:
> - Abstract: The authors mention several times the possibility to run high resolution simulations. What is the targeted resolution? Do the scheme need any modification for this high resolution?

The upper limit of the target resolution is given by the resolution of the updated input data. The target will be T255 (about 0.5 degree at the equator) or higher (which so far is only mentioned in the conclusions). With at least 0.1 degree resolution, the new input data will also serve considerably higher resolving simulations. Also the emission scheme itself can be used straight forwardly at higher resolution. As it is not entirely resolution independent the overall scaling might need to be adjusted. We have added the following sentence to the introduction:

Page 2, line 29 "To equip the model for simulations at resolution T255 (about 0.5 degree) or higher, new input data should have at least 0.1 degree resolution."

And in section 3:

Page 6, line 7 "When switching to different model resolutions, the scaling factor can be used to balance potential resolution dependencies of the emission scheme."

> - Page 2 Lines 18-19: The authors should justify the "rapid changes of deserts and semi-arid regions in recent decades"

References to Figs. 1 and 2 and literature have been included.

> • Page 3 Section 2.1: Looking at Fig.1, I get the impression that there are more regions with shrinking deserts, is it true?

That is correct, the area with positive correlation coefficient covers $1.3 \cdot 10^6$ km$^2$ globally which is about half the area with negative correlation coefficient ($2.6 \cdot 10^6$ km$^2$). Additionally, the regions of shrinking deserts are spread over a larger area because they are predominantly surrounding the large deserts whereas expanding source areas are located more centrally. We have added the numbers to the text.

> • Page 3 Line 27: Any justification for the equation (1) giving the vegetation factor? Is it used in other models?

The vegetation factor is the same as used by Astitha et al. 2012 and interpolates linearly between full emissions for no vegetation and entirely suppressed emissions for LAI > 0.35; the threshold value was introduced by Mahowald et al. 1999. We have added the references.

> • Page 3 Lines 29-30: Could the authors clarify which statistical test they have used?

The trend has been calculated for each pixel by fitting a linear regression model to the time series of annual average LAI values using least squares. The resulting slope yields the trend and is considered significant (i.e., it is plotted in Fig. 2) if the corresponding p value is below the significance level of 0.05. We have rephrased the sentence.

> • Page 4 Section 2.3: Contrary to Sections 2.1 and 2.2, the authors have not elaborated on the differences between the two versions of the clay fraction maps. Which is the expected impact on dust emissions?

The two versions of the clay fractions are now compared in Figure S1 in supplement. The impact on the dust emissions is discussed in the new section 4. The expected impact of the new data is a better representation of details below the 1 degree resolution such as river valleys. Moreover, as mentioned, the new data is more appropriate to represent the relevant topmost soil layer. As the clay fraction map is assumed to be static (based on the longer typical time-scales of the relevant geological processes), unlike in Sect. 2.1 and 2.2 we could not perform trend and variation analysis.

> • Page 4 Lines 26-29: Is there any work forecast to include again the effect of soil moisture on dust emissions? It might be important in some regions like Sahel.

We agree that the effect of soil moisture is important in regions like the Sahel (or the Middle East) and it would be very desirable to consider it in the model. A prerequisite would be a soil model more detailed than the current bucket model to obtain soil moisture values for only the topmost surface layer. While the inclusion of new soil models in EMAC is discussed, to our knowledge one has yet to be implemented.

> • Page 5 Line 5: This equation differs from the one given in Astitha et al. (2012), the authors should correct it or explain why it is different.

Eq. (9) used by Astitha et al. (2012) implies that the horizontal flux H is proportional to

$$u_*^3(1 + u_{*t}/u_*)(1 - u_{*t}^2/u_*^2) = u_*^3(1 + u_{*t}/u_*)(1 + u_{*t}/u_*)(1 - u_{*t}/u_*)$$
$$= (u_* + u_{*t})(u_* + u_{*t})(u_* - u_{*t})$$

which agrees with the RHS of our Eq. (2).

> • Page 5 Line 9: The authors should justify the choice of 0.4 m/s, and clarify what they call "good results" explaining what has been compared.

The value is justified by the results presented in Sect. 4 which are good in the sense that the validation simulation produces, compared to observations, significantly more realistic results (in terms of skill scores and correlation coefficients) than the reference simulation using the original emission scheme of Astitha et al. (2012)

which already proofed to yield realistic results in other studies. We have added results from simulations using different limits in the supplement (Figure S3).

> - Page 5 Line 29: The parameter dmax could be added in Table 2

Dmax has been added to the table.

> - Page 6 Lines 10-15: I did not understand if finally the chemical composition of dust is included or not in the model.

As mentioned on Page 6, lines 31f the chemical composition is included in the model for both, reference and validation run. As the corresponding changes to the dust emission scheme are independent of all other modifications, do not affect total dust emission flux and can be used with the original and the revised emission scheme, their effects (see Karydis et al. 2016) have been excluded from the evaluation, but code and data are released with the revision presented here.

> - Page 6 Lines 19-21: The list of submodels is unclear for readers not familiar to the model. The authors should add a reference to have the details about these parameterizations.

We have added a reference (http://www.messy-interface.org/current/auto/messy_submodels.html).

> - Page 6 Line 25: What is the Tanré climatology used for? (AOD or only other optical properties?)

It is used for extinction, single scattering albedo and asymmetry factor (now mentioned in the text).

> - Page 6 Line 30: A reference to Table 1 should be added to present the simulations.

The reference has been added.

> - Page 6 Line 32: Is a one-year simulation long enough to evaluate the revised dust emissions? Is there any reason to select the year 2011?

While longer simulations are preferable, the one-year period suffices to yield statistically significant differences between reference and validation simulation and has been chosen considering the computationally expensive model setup used. The year 2011 has been selected to represent a recent period well past the time period on which the old, outdated input data is based on, and to allow to continue the simulation within the period of available new input data in case this would have been necessary to collect more statistics.

> - Page 7 Line 15: Which level of AERONET AOD has been used in this comparison?

Level 2 data has been used (now mentioned in the text).

> - Page 7 Line 17: Maybe the authors should divide the region B in two sub-regions, for the reader to identify more easily the different stations.

We have divided the region into a northern and southern part.

> - Page 7 Lines 29-30: I don't understand how this skill score based on correlation can be affected by a bias.

The overestimation of the AOD during dust events results in an overestimated amplitude of the AOD variation between dust-free (when both reference and validation simulation yield AODs close to zero) and dusty periods. Accordingly, variance and standard deviation are overestimated, the latter entering the skill score defined in Eq. (5).

- Page 7 Section 4.1: It could be also useful to add one or two time series in stations where the score has increased.

We have added time series plots of the five stations with the largest increase to the supplement (Fig. S5).

- Page 8 Line 1: Which is the altitude of the model grid cell?

The model grid cell has a surface altitude of 63 m which we now mention in the text.

- Page 8 Line 21: Is this increase of spatial correlation statistically significant?

The statistical error of the numbers is small, the error estimates obtained by jackknife resampling of the more than $10^5$ pixel values are 0.004 for the correlation coefficients and 0.006 for the skill scores. Therefore, the digits provided are presumably exact and the probability for the increase under the null hypothesis (assuming no improvement) is virtually zero.

It should be stressed that the improvements reflected by the numbers are substantial and the improvement of the global AOD distribution is a major advantage of the revised emissions.

- Page 8 Lines 28-30: The authors could think about adding a score for the measuring the improvement in seasonal cycle, which could reinforce the robustness of their results.

We have added statistical analysis of the seasonal AOD and DAOD values over the Arabian Peninsula to the supplement (Figs. S6, S7).

- Page 9 Line 17: Same remark for the significance of the increase in the skill score.

Here, the error estimates for the reference results are slightly larger than above (0.012 for the correlation coefficient, 0.017 for the skill score) due to the distinct peaks in the reference DAOD distribution, but still very small compared to the increase due to the revised emissions, therefore again the increases are highly significant.

We apologise that the numbers in the text (page 9, lines 16f) are not the correct numbers provided in the caption of Figure 12. This has been fixed.

- Page 9 Line 32: The time dependence of land cover and vegetation has not been tested here because the simulations were too short.

We agree, however, using the input data based on observations from the year simulated (2011) likely contributed to the more realistic results.

Technical comments:
- Page 2 Lines 8-9: The abbreviations DU_Astitha1 and DU_Astitha2 are useless since they are not used in the rest of the paper.

The abbreviations are the names of the corresponding options in the EMAC setup. To unambiguously specify the emission scheme our study builds on, we would like to keep mentioning the names here.

- Page 6 Line 24: ISORROPIA

The typo has been fixed.

- Figure 1: The color bar should be changed, because the values below -0.2 cannot be distinguished.

The contrast has been increased.

- Figure 8: The authors could replace the letters (A, B, etc.) by the name of the regions in the blue line at the top of the figure.

We have introduced more descriptive abbreviations.

- Figure 9: AERONET data is represented with dots in the figure, while it is a line in the caption.

This has been fixed.

- References: The format needs to be homogenized (notably the use of first names for the first author).

The bibliography has been revised.

---

## Author Comment (AC2) · 17 Nov 2017

**Reply to referee 2**

We thank the referee for the thorough review and the helpful advices to improve the article. They have all been considered in a revised manuscript and a new supplement. In the following please find our replies to the individual comments.

> General comments This study presents updates of the dust emission scheme implemented in the global atmospheric chemistry model EMAC based on the previous work of Astitha et al. (2012). The land cover, vegetation topography and clay fraction maps are updated to more recent versions using higher spatial resolution. Changes are also imposed to the dust emission scheme directly. The updated dust emissions are evaluated with AOD measurements from AERONET, MODIS and IASI for the year 2011. The title, flow and structure of the paper are appropriate. All updates are well received and long needed, given the importance of quality input data to accurately parameterize physical processes that cannot be described by first principles. However, the authors keep the evaluation part largely on the qualitative side, which does not help the reader and the community to fully understand why these changes were impactful and significant. The conclusions are also very brief for a model development/improvement paper. The authors miss a great opportunity to discuss the very interesting aspects of each revision and inform the community of which one should be considered more impactful (if not all). The specific comments below will help the authors revise the paper so it can be accepted for publication with GMD.

We have added a section discussing the effects of the individual modifications. The evaluation - comparing numeric results for AOD, DAOD, correlation coefficients and skill scores - has been extended by even more quantitative comparisons of dust concentrations and deposition rates.

> Specific comments/suggestions
> Section 2.2 (Vegetation): Please elaborate on the calculation of fveg (Eq.1): What is the role of 0.35 and what is the meaning of fveg being 1 or less than 1.

This vegetation factor, also used by Astitha et al. 2012, interpolates linearly between full emissions for no vegetation and entirely suppressed emissions for LAI > 0.35 which was introduced as threshold by Mahowald et al. 1999. We have added this information.

> Section 2.3 (Clay fraction): provide a map of the updated clay fraction in comparison to the one previously used in the model. It will provide context on the significance of changes that later affect the parameterization scheme.

The map has been added to the supplement.

> Section 3 Page 4 (soil moisture): The soil moisture term in Astitha et al. (2012) and Eq. A1 in this paper is omitted from the threshold friction velocity. However, the authors correctly describe the dependence of dust emission on soil moisture at the end of this paragraph. What is not clear is if the statement "we consider a detailed parametrisation of the soil moisture effect to be essential to capture the observed trends in future simulations. This will require a comprehensive soil model providing accurate moisture values for the topmost surface layer" refers to an action already taken for this study or a future goal. In any case, a discussion on how the exclusion of soil moisture correction influences the simulations is important here.

The more comprehensive soil model is a future goal, we have clarified the statement and added a figure illustrating the effect of omitting the factor to the supplement.

> Page 5, (Surface friction velocity limit): a note must be placed that Eq.2 holds only when u* > u*t. Also, choosing to limit the threshold velocity to a maximum value of 0.4 m/s seems arbitrary and needs to be elaborated. What led the authors to this specific value? Some context and rationale must be provided.

We have clarified the equation and included results for different limits in the supplement.

Page 5 (Topography factor): I am not sure of the role of the normalization factor 5.3 and how it conserves global emissions. It sounds like a tuning factor to me, so please elaborate on the role of the factor and the method used to estimate it.

Using the topography factor $S_{\text{topo}}$ as given in Eq. (3) has two effects: the desired effect is that it adjusts the spatial distribution of the emissions, but since by definition $0 \leq S_{\text{topo}} \leq 1$ and usually $S_{\text{topo}} < 1$ an undesired side effect is the reduction of the emissions globally. We quantified this reduction in a one month simulation obtaining a ratio between the global emissions without and including the factor $S_{\text{topo}}$ of 5.3. Consequently, we include $5.3 \times S_{\text{topo}}$ instead of just $S_{\text{topo}}$ and thereby conserve the global emissions. In practice, this normalisation factor can be combined with the empirical scaling factor c, hence it introduces no additional tuning factor. We have expanded the corresponding corresponding text.

Page 5 (Mode mapping): This is not a strict update of the emission scheme but rather an alteration in order to use the GMXE aerosol model compared to M7 used by Astitha et al. (2012). A brief note must be included in this section to clarify that the original scheme (as well as the reference simulation herein) used a different aerosol module thus a different approach to particle size distribution. The omission of the eight transport size bins is surely a change from the original version.

In this study, we use the GMXE submodel for both, reference and validation simulation. Since GMXE is based on M7 and uses the same modal concept, the question of how to map the three emission modes to the aerosol submodel modes is unaffected by this choice. Since in the original scheme the dust was not further processed while in the "transport" bins but directly mapped to the GMXE/M7 modes, skipping this step is in fact an implementation detail and when aligning the threshold between accumulation and coarse mode with the bin boundary at radius 0.6 um yields identical results.

Figure 6: what is the higher value in this scale (above 0.1)? When we see 0.1 fraction of Ca++, does that indicate the mass, volume (or else) fraction of the total particles within each specific grid cell? A better explanation of the mineral cations fraction could be included also in page 6 (last paragraph of section 3).

The upper limit of the scale is 0.12, which is reached by the Ca++ fraction in the Kalahari and Taklamakan Desert; we have added a tick mark. The fractions shown in Figure 6 are mass fractions; we have added this missing information in the caption.

Section 4.1, page 7: 1. "On the other hand, dust events observed by AERONET in January and December are reproduced by the validation simulation, but not by the reference simulation": this comparison is not at all discernible in the plot as it is. If this is an important argument, the plot must be revised somehow to make the statement visible.

We have marked the events we are referring to in the plot.

2. Given that Izana and La Laguna are within the same model grid cell, an average of the AOD from both sites could be an alternative way to compare with the model value. In addition, when evaluating numerical model simulations one can employ the nearest neighbor (as done here) or a bilinear interpolation between the observation and the model value from the four closest grid points.

Averaging the AOD values of all stations within the same grid cell generally is a reasonable strategy, in this case, however, even La Laguna station at 568 m altitude is not representative for the grid cell which is mostly covered by ocean. Since the neighbouring cells are also predominantly covered by water, bilinear interpolation would not make a big difference in this regard. Better agreement could be obtained by computing the model AOD at station altitude rather than at model surface height. Generally, such distinctive sub-grid topographies and shore lines reveal the limitations of the model resolution.

> 3. The main criticism I have for the evaluation using the skill score (as with any other statistical metric) is the qualitative determination of which configuration provided the best results. Characterizations such as "slightly better" or "marginally larger" do not show robustness in the performed evaluation. My immediate question is: are these differences statistically significant? Are they statistically different? This is the only way to prove or convince the audience that a, say, 0.05 change in the skill score is significant enough.

Measures like the skill score are supposed to quantify agreement. The significance of the skill score improvement has only been indicated by the dominance of green bars in Figure 8. In the revised manuscript we have amended the figure to depict error estimates for the $\Delta S$ values.

> 4. What about using AOD of coarse vs. accumulation modes from AERONET?

The fine/coarse mode AOD product is more sparse than the AOD data, moreover it is only available at 500 nm and we do not have corresponding model output available. We will extend the evaluation to other observables instead (see below).

> 5. How about using total PM concentrations wherever available (and for cases of high dust concentrations) to evaluate model performance? An additional means of quantitative evaluation needs to be included.

The evaluation will be extended by comparisons with dust concentration and deposition data.

> Figure 8: Please consider replacing "Regions A, B," etc. from the figure with the names of the regions as it is not convenient to go back and forth between Fig. 7 and 8 to identify the regions.'

A more convenient naming has been introduced.

> Sections 4.2 and 4.3: The scarcity of desert dust concentration measurements is a well-known problem in the modeling community when assessing model and parameterization scheme performances. This is when satellite and remote sensing observations come into play and are important tools of assessment. Nevertheless, leaving the comparison in the qualitative state only, influences the robustness of the conclusions. Looking at the IASI zoomed plots (Fig. 13), I would not immediately say that the validation is better than the reference simulation. They are different and somewhat both incorrect in my view. If the authors presented a quantitative assessment of the performance, there would be no doubt on the comparison. Also, why is the zoomed area over Middle East only? What is the special interest for this specific region? This has to be explained thoroughly so it will not be seen as "cherry picking".

To guide the eye in Figs. 11 and 13 we have marked the region where we see considerable improvements, which is the Arabian Peninsula including Iraq, Syria and Jordan. To corroborate the improvements, we have evaluted the spatial correlations and skill scores in this region which significantly increase as shown in the new Figs. S6 and S7 in the supplement. The Middle East is of special interest because there the original emission scheme clearly suffered from outdated input data, as mentioned in the introduction. To avoid the impression of cherry picking we now provide seasonal global plots in a supplement.

> Conclusions: the conclusions are quite brief (two sentences in the end of the section). More discussion should be invoked on how the changes influence a better model performance (as long as there is a robust determination of "better" or "worse"). There are very interesting aspects in this study and it would be very useful for the community to understand how the changes that were implemented individually affect dust emissions. I believe that adding such discussion would greatly strengthen the paper.

The conclusions now reflect the additional aspects covered by the revision.

Appendix: There are a couple of things missing in the depiction of the emission flux jemis (Eq A2): 1) I don't see the mass fraction (source to transport bins, M in Eq.4) that should be multiplied in the right side of the equation. 2) In Astitha et al. (2012) they used the relative surface area covered from particles with diameter D (Srel) to calculate the horizontal flux H. Did the authors omit this calculation in their revisions?

The emission flux given in Eq. (A2) is the total emission flux. In the revised manuscript we have multiplied the mass fraction to present the flux for each mode instead. The factor $S_{\text{rel}}$ is only utilised in the emission scheme variant DU_Astitha2, not in the scheme DU_Astitha1 used in this study.

---

## Author Comment (AC3) · 17 Nov 2017

We propose to amend the title to "Revised mineral dust emissions in the atmospheric chemistry-climate model EMAC (MESSy 2.52 DU_Astitha1 KKDU2017 patch)" and will discuss this suggestion with the responsible editor. The pre-formulated lines on the messy interface homepage have been included in the code availability section. The corresponding author acts as point of contact to obtain code and data as long as they are not yet included in an official MESSy release and the common data pools (the code will become part of the official MESSy code as soon as this manuscript is published).

---

## Author Response (AR2)

**Changes**

On the  following pages the two minor changes made since the last file upload are highlighted.

**Revised mineral dust emissions in the atmospheric chemistry-climate model EMAC (MESSy 2.52 DU_Astitha1 KKDU2017 patch)**

Klaus Klingmüller[1], Swen Metzger[2,4], Mohamed Abdelkader[1,3], Vlassis A. Karydis[1], Georgiy L. Stenchikov[3], Andrea Pozzer[1], and Jos Lelieveld[1,2]

[1]Max Planck Institute for Chemistry, P.O. Box 3060, 55020 Mainz, Germany

[2]The Cyprus Institute, P.O. Box 27456, 1645 Nicosia, Cyprus

[3]King Abdullah University of Science and Technology, Thuwal 23955-6900, Saudi Arabia

[4]ResearchConcepts io GmbH, Freiburg im Breisgau, Germany

*Correspondence to:* Klaus Klingmüller (k.klingmueller@mpic.de)

**Abstract.** To improve the aeolian dust budget calculations with the global ECHAM/MESSy atmospheric chemistry-climate model (EMAC) we have implemented new input data and updates of the emission scheme.

The data set comprises landcover classification, vegetation, clay fraction and topography. It is based on up-to-date observations, which is crucial to account for the rapid changes of deserts and semi-arid regions in recent decades. The new Moderate-resolution Imaging Spectroradiometer (MODIS) based landcover and vegetation data is time dependent, and the effect of long-term trends and variability of the relevant parameters is therefore considered by the emission scheme. All input data has a spatial resolution of at least $0.1°$ compared to $1°$ in the previous version, equipping the model for high resolution simulations.

We validate the updates by comparing the aerosol optical depth (AOD) at 550 nm wavelength from a one year simulation at T106 (about $1.1°$) resolution with Aerosol Robotic Network (AERONET) and MODIS observations, the 10 $\mu$m dust AOD (DAOD) with Infrared Atmospheric Sounding Interferometer (IASI) retrievals, and dust concentration and deposition results with observations from the AEROCOM dust benchmark data set. The update significantly improves agreement with the observations and is therefore recommended to be used in future simulations.

Also the comparison with dust deposition observations shows improved agreement when using the updated emissions. This is less clear for the comparison with dust concentration data, where original and updated emission scheme do not show a significant performance difference.

5 While the updates clearly improve the global distribution of aeolian dust, the total amount of globally emitted dust remains unchanged and consistent with literature values.

Subject to the future availability of suitable soil models in EMAC providing soil moisture values for a thin surface soil layer, the activation of the explicit soil moisture dependency of the threshold surface friction velocity might further improve the agreement with observed trends and variability.

**Code and data availability**

10 The Modular Earth Submodel System (MESSy) is continuously further developed and applied by a consortium of institutions. The usage of MESSy and access to the source code is licenced to all affiliates of institutions which are members of the MESSy Consortium. Institutions can become a member of the MESSy Consortium by signing the MESSy Memorandum of Understanding. More information can be found on the MESSy Consortium Website (http://www.messy-interface.org). The input data files and all modifications to the EMAC source code presented in this article are available on request until they 15 become part of the official MESSy code.

*Acknowledgements.* The research reported in this publication has received funding from the King Abdullah University of Science and Technology (KAUST) CRG3 grant URF/1/2180-01 *Combined Radiative and Air Quality Effects of Anthropogenic Air Pollution and Dust over the Arabian Peninsula* and was supported by the European Space Agency as part of the Aerosol_cci project. S. Metzger received funding from the European Commission through the H2020-EINFRA-2015-1 project "Energy oriented Centre of Excellence for computer 20 applications (EoCoE)", Proposal number: 676629.